

**A Machine Learning-Based Cloud Detection and Thermodynamic**
**Phase Classification Algorithm using Passive Spectral Observations**
**Chenxi Wang[1,2], Steven Platnick[2], Kerry Meyer[2], Zhibo Zhang[3], Yaping Zhou[1,2]**
[1]Joint Center for Earth Systems Technology, University of Maryland Baltimore County,
Baltimore, MD, USA
[2]NASA Goddard Space Flight Center, Greenbelt, MD, USA.
[3]Department of Physics, University of Maryland Baltimore County, Baltimore, MD, USA.





**Abstract**
We trained two Random Forest (RF) machine-learning models for cloud mask and cloud
thermodynamic phase detection using spectral observations from VIIRS on Suomi-NPP (SNPP).
Observations from CALIOP were carefully selected to provide reference labels. The two RF
models were trained for all-day and daytime-only conditions using a 4-year collocated
VIIRS/CALIOP dataset from 2013 to 2016. Due to the orbit difference, the collocated CALIOP
and SNPP VIIRS training samples cover a broad viewing zenith angle range, which is a great
benefit to overall model performance. The all-day model uses 3 VIIRS infrared (IR) bands (8.6,
11, and 12 $\mu$m) and the daytime model uses 5 Near-IR (NIR) and Shortwave-IR (SWIR) bands
(0.86, 1.24, 1.38, 1.64 and 2.25 $\mu$m) together with the 3 IR bands to detect clear, liquid water, and
ice cloud pixels. Up to 7 surface types, namely, ocean/water, forest, cropland, grassland, snow/ice,
barren/desert, and shrubland, were considered separately to enhance performance for both models.
Detection of cloudy pixels and thermodynamic phase with the two RF models were compared
against collocated CALIOP products from 2017. It is shown that the two RF models have high
accuracy rates in comparison with the CALIOP reference for both cloud detection and
thermodynamic phase. For cloud detection, the accuracy rates of the daytime RF model are higher
than 92% for all surface types, while the accuracy rates of the all-day RF model decrease by 3~8%,
depending on surface type. For cloud thermodynamic phase, both RF models agree well with
CALIOP, except over barren/desert regions. Other existing SNPP VIIRS and Aqua MODIS cloud
mask and phase products are also evaluated, with results showing that the two RF models and the
MODIS MYD06 optical property phase product are the top 3 algorithms with respect to lidar
observations during the daytime. During the nighttime, the RF all-day model works best for both
cloud detection and phase, in particular for pixels over snow/ice surfaces. The present RF models
can be extended to other similar passive instruments if training samples can be collected from



CALIOP or other lidars. However, the quality of reference labels and potential sampling issues
that may impact model performance would need further attention.

**1. Introduction**

Detection and classification (DC) of atmospheric constituents using satellite observations is
often a critical initial step in many remote sensing algorithms. For example, a prerequisite for cloud
optical and microphysical property retrievals is identifying the presence of clouds, i.e., a
clear/cloudy classification [*Frey et al.*, 2008]. Additionally, characteristics such as cloud
thermodynamic phase are needed as they can strongly impact the scattering/absorption properties
of cloud droplets/particles [*Platnick et al.*, 2017]. Similarly, current operational aerosol algorithms
can only retrieve aerosol optical depth (AOD) for "non-cloudy" pixels since even slight cloud
contamination can result in erroneously high retrieved AOD [*Remer et al.*, 2005]. Therefore, errors
in detecting and classifying atmospheric components can significantly impact downstream
retrieval products and scientific analyses.
There are many examples of traditional DC algorithms designed for satellite instruments. For
example, the Moderate Resolution Imaging Spectroradiometer (MODIS) has algorithms
developed for cloud masking [*Frey et al.*, 2008; *Ackerman et al.*, 2008], cloud thermodynamic
phase [*Baum et al.*, 2012; *Marchant et al.*, 2016], aerosol type [*Levy et al.*, 2013; *Sayer et al.*,
2014], and snow coverage over land surfaces [*Hall and Riggs*, 2016]. Decision trees or voting
schemes involving multiple thresholds are typically used in these traditional algorithms. The
decision tree branches, tests, and thresholds are often determined empirically after a tedious hand
tuning/testing process based on the developer's experience and access to validation datasets.
Further, the branches and thresholds are often very sensitive to the specific instrument (e.g.,
spectral band pass, calibration, noise characteristics, view/solar geometry sampling). Therefore,
an obvious weakness of these traditional methods (e.g., decision trees/voting schemes/thresholds)



is that it is challenging and time consuming to develop algorithms across multiple instruments and
to maintain performance for a single instrument having radiometric stability issues. Meanwhile, a
well-designed traditional method may have remarkable performance in a specific region and
season yet have significant biases when applied globally and/or annually [*Cho et al.*, 2009; *Liu et*
*al.*, 2010; *Zhou et al.*, 2019]. Additional complexities arise when DC problems become more non-
linear across large spatial and temporal scales, and more variables need to be considered. It is
difficult to develop and apply a single or a few decision trees to complicated non-linear problems
that are controlled by dozens or more variables. As expected, a single decision tree can grow very
deep and tend to have a highly irregular structure in order to consider a large number of features
(variables) simultaneously, leading to a significant overfitting effect (i.e., an over-constrained
training that makes predictions too close to the training dataset but fails to predict future
observations reliably). For example, MODIS provides an all-day cloud phase product based only
on infrared (IR) observations (hereafter referred to as IR-Phase [*Baum et al.*, 2012]). Although it
can be expected that the tests and thresholds should vary with satellite viewing geometry [*Maddux*
*et al.*, 2010], full consideration of viewing geometries, together with the variations of many other
factors such as surface emission, geolocation, and cloud properties, is very challenging based on
manual tuning. As a consequence, it is found that the liquid water and ice cloud fractions from the
IR-Phase product exhibit noticeable view zenith angle (VZA) dependency (see Figure 12). This is
an undesirable but unavoidable artifact since cloud phase statistics should be independent from
solar/viewing geometry. Such VZA dependencies may strongly affect similar products from
geostationary instruments because of the fixed VZA-geolocation mapping. Similar artifacts may
also impact aerosol type and retrieval products [*Wu et al.*, 2016]. Finally, it is difficult to acquire
pixel-level classification uncertainties with traditional methods.



In contrast to traditional methods, Machine Learning (ML) based DC algorithms are designed
to autonomously find information (e.g., patterns of spectral, spatial, and/or time series) in one or
more given datasets and learn hidden signatures of different objects. An obvious advantage of ML
models is that the training process is efficient and highly flexible. Manually defined thresholds or
matching conditions to expected spectral patterns are no longer needed. In this paper, we developed
two ML-based DC algorithms for detecting cloud and cloud thermodynamic phase for different
local times (i.e., daytime and nighttime) with observations from the Visible Infrared Imaging
Radiometer Suite (VIIRS) on Suomi-NPP (SNPP). The ML models are trained with collocated
observations from SNPP VIIRS and Cloud-Aerosol Lidar with Orthogonal Polarization
(CALIOP), with CALIOP data used as reference. In Section 2, we give a brief discussion of the
ML models. Data generated for model training and validation will be introduced in Section 3.
Details of the model training and evaluation are shown in Section 4. Section 5 discusses the
advantages and potential limitations of the present ML models. Conclusions are given in Section

94      6.

**2. Traditional DC methods and Machine Learning Models**
**2.1 Traditional DC methods**
All DC algorithms with remote sensing observations are based on the underlying physics of
the spectral, spatial, and/or temporal structures of specified objects. In traditional DC algorithms,
all the physical rules and structures have to be explicitly defined as various tests and thresholds.
For example, the MODIS MOD35/MYD35 cloud mask algorithm uses more than 20 tests with
visible/near-infrared (VNIR), shortwave-infrared (SWIR), and infrared (IR) observations [*Frey et*
*al.*, 2008] that are carefully designed to consider numerous scenarios, including different surface
types (e.g., ocean, land, desert, snow, etc.) and local times (day/night). Similar algorithms are
designed for aerosol type and cloud thermodynamic phase classifications. As an example, Figure



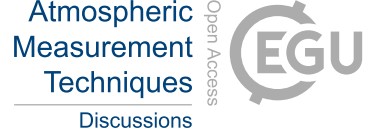

1 illustrates spectral patterns of 5 typical daytime oceanic scenes (pixel types) observed by SNPP
VIIRS. Spectral pattern of each of the 5 scenes, namely, clear sky, liquid water cloud, ice cloud,
dust, and smoke, is averaged by using more than 1,000 pixels with the same type. It is clear that
the 5 scenes are different in either reflectance ratios between a given VNIR/SWIR band and the
0.86 $\mu$m band, or brightness temperature differences (BTD) between two IR window bands.
Consequently, such spectral features are frequently used to differentiate pixel types in DC
algorithms. In addition to spectral patterns, simple methods are developed to take into account
spatial information. For example, it is found that cloud reflectance usually has larger spatial
variability than aerosols [*Martins et al.*, 2002] and clear sky pixels [*Platnick et al.*, 2017].
Therefore, spatial variabilities of VNIR and SWIR reflectance bands are used to differentiate
clouds from non-cloudy pixels in the current MODIS clear sky restoral (CSR) algorithm [*Platnick
et al.*, 2017] and Dark Target aerosol retrieval algorithm [*Levy et al.*, 2013].
**2.2 Machine learning models**
Different from the traditional DC methods, ML algorithms are developed to autonomously
learn the hidden spectral/spatial/temporal patterns of different objects. Consequently, manually
defined thresholds or matching conditions to expected patterns are no longer needed. In image
recognition applications, numerous ML algorithms [e.g., *Joachims* 1998; *Breiman* 1999;
*Dietterich* 2000] have been developed for independent pixels using a single or small number of
decision trees in late 1990s. *Ho* [1998] and many other studies have demonstrated that although
these single or small number of decision trees can always provide maximum prediction accuracies
in training processes, significant overfitting effects cannot be avoided. Tremendous efforts have
been made to overcome the dilemma between maintenance of prediction accuracy and avoiding
overfitting.   Among these, the Random Forest (RF) and Gradient Boosting (GB) algorithm
[*Breiman* 1999; *Dietterich* 2000; *Friedman* 2001] provide a framework of using a large number of
decision trees (ensemble) but a subset of features in each tree to achieve optimization in the
performance. It has been proven that the ensemble-based algorithms can largely correct mistakes
made by individual trees [*Ji and Ma*, 1997; *Tumer and Ghosh*, 1996; *Latinne et al.*, 2001] and
avoid overfitting [*Freund et al.*, 2001]. Currently, the RF and GB algorithms are frequently used
in non-linear classification and regression problems. For example, RF models have been used in
several cloud/aerosol remote sensing applications, such as differentiating cloudy from clear
footprints for the Clouds and the Earth's Radiation Energy System (CERES) instrument [*Thampi*
*et al.*, 2017], estimating surface-level PM2.5 concentrations [*Hu et al.*, 2017], and detecting low
cloud using Advanced Baseline Imager (ABI) on Geostationary Operational Environmental
Satellites (GOES) [*Haynes et al.*, 2019]. In our study, we also choose the RF model based on its
proven record in earth science applications.
In the RF model, a final prediction is made based on majority vote computed from probability
($P_i$) of each class ($i^{th}$):

$$P_i = \frac{w_i N_i}{\sum_{j=1}^{j=m} w_j N_j},$$  (1)

where $m$ is the total number of classes, $N_i$ and $N_j$ are the number of trees that predict the $i^{th}$ and $j^{th}$
classes, and $w_i$ and $w_j$ are weightings for the $i^{th}$ and $j^{th}$ classes, respectively. If all trees are equally
weighted, $w$ for individual classes are equal to 1. The two most important parameters for tuning
the RF algorithm are the number of decision trees ($N_{Tree}$) and the maximum tree depth ($N_{Depth}$).
However, an optimal definition of these two parameters is still an open question [*Latinne et al.*,
2001]. Larger $N_{Tree}$ and $N_{Depth}$ provide more accurate predictions, at the cost of significantly
increased computational resources. For many cases, larger $N_{Depth}$ may cause overfitting effects
[*Oshiro et al.*, 2012; *Scornet*, 2018]. Generally, the two parameters have to be large enough to let
the decision trees have a relatively wide diversity and capture the hidden patterns. For practical





purposes, however, the two parameters cannot be too large to prevent the models from overfitting
and to reduce computing burden [*Latinne et al.*, 2001; *Scornet* 2018].
In this study, we adopt a widely applied RF algorithm in the Scikit-learn Machine Learning
package [*Pedregosa et al.*, 2011]. We train two RF models for object DC using SNPP VIIRS
spectral observations at two observational times: an all-day RF model using three VIIRS thermal
IR observations (hereafter referred to as the RF all-day model) and a daytime-only RF model that
uses both VNIR/SWIR and thermal IR observations (hereafter the RF daytime model). The models
are trained to detect clear sky, liquid water cloud, and ice cloud pixels with single pixel level
information. Parameters of the two RF models will be tuned and tested carefully to achieve the
best accuracy and to avoid the overfitting effect. Details will be discussed in Section 4.
**3. Data**
**3.1 Reference label of pixels**
Space-borne active sensors, such as CALIOP onboard CALIPSO [*Winker et al.*, 2013], the
Cloud-Aerosol Transport System (CATS) [*McGill et al.*, 2015] onboard the International Space
Station (ISS), and CPR on board CloudSat [*Stephens et al.*, 2002], are frequently used to evaluate
the performance of traditional cloud/aerosol DC and property retrieval algorithms designed for
passive sensors [*Stubenrauch et al.*, 2013; *Wang et al.*, 2019]. Until its exit on 13 September 2018
(to join CloudSat in the C-Train), CALIPSO was a key member of the Afternoon Constellation of
satellites (A-Train), and began providing profiling observations of the atmosphere in 2006 [*Winker*
*et al.*, 2013]. The CALIPSO lidar CALIOP operates at wavelengths of 532 nm and 1064 nm,
measuring backscattering profiles at a 30-meter vertical and 333 m along-track resolution.
CALIOP also measures the perpendicular and parallel signals at 532 nm, along with the
depolarization ratio at 532 nm that is frequently used in cloud phase discrimination algorithms

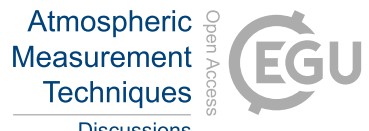

because of its strong particle shape dependence. The CALIOP Version 4 Level 2 1 km Cloud Layer
product will be used to provide reference cloud phase labels in both model training and validation
stages.
While the CATS lidar and the CloudSat radar CPR also provide profiling information, both
have limitations that preclude their use here. CATS had a relatively short life time (from January
2015 to October 2017), and its low inclination angle (51°) orbit aboard the ISS excludes sampling
of high-latitude regions [*Noel et al.*, 2018]. CloudSat CPR observes reflectivity profiles at 94-GHz,
which are more sensitive to optically thicker clouds consisting of large particles but are blind to
aerosols and optically thin clouds. CloudSat also has difficulty in detecting clouds near the surface
due to the surface clutter effect [*Tanelli et al.*, 2008]. Therefore, only CALIOP data are used to
provide reference cloud phase labels in this study.
**3.2 RF model input**
It should be pointed out that ML models use similar input datasets as traditional methods. The
input variables (features) and reference labels of the present RF models are carefully selected based
on prior physical knowledge of the spectral characteristics of each object.
The SNPP VIIRS and the NOAA-20+ series provides spectral observations from 0.4 to 12 $\mu$m
at sub-kilometer spatial resolutions [*Lee et al.*, 2006]. Specifically, VIIRS has 16 moderate
resolution bands (M band) and 5 higher resolution imagery bands (I band) at 750 m and 375 m
nadir resolutions, respectively. The spectral capabilities of VIIRS allow for extracting abundant
information on the surface and atmospheric components, such as clouds [*Ackerman et al.*, 2019]
and aerosols [*Sayer et al.*, 2018]. It is also worth noting that VIIRS utilizes an on-board detector
aggregation scheme that minimizes pixel size growth in the across-track direction towards swath
edge [*Cao et al.*, 2013]. As an example, although the VIIRS M-bands and MODIS 1 km bands



have similar nadir spatial resolutions, the VIIRS across-track pixel size increases roughly only to
1.625 km at scan edge, which is much smaller than the MODIS pixel size of roughly 4.9 km at
scan edge [*Justice et al.*, 2011]. Another obvious advantage of using SNPP VIIRS rather than Aqua
MODIS data is that, due to the CALIPSO and SNPP orbit differences, the training samples cover
a broader viewing zenith angle range, which is a great benefit to overall model performance.
Consequently, Level-1B M-band observations from the SNPP VIIRS are used here.
Ancillary data, including the surface skin temperature, spectral surface emissivity, surface
types, and snow/ice coverage, are important in cloud DC related remote sensing applications [*Frey*
*et al.*, 2008; *Wolters et al.*, 2008; *Baum et al.*, 2012] and cloud/aerosol retrievals [*Levy et al.*, 2013;
*Wang et al.*, 2014; 2016a; 2016b; *Meyer et al.*, 2016; *Platnick et al.*, 2017]. The inst1_2d_asm_Nx
product (version 5.12.4) from the Modern-Era Retrospective Analysis for Research and
Applications, Version 2 (MERRA-2) [*Gelaro et al.*, 2017] is utilized to provide the hourly
instantaneous surface skin temperature and 10-meter surface wind speed. The UW-Madison
baseline fit land surface emissivity database [*Seemann et al.*, 2008] and the Terra/Aqua MODIS
combined Land surface product (MCD12C1 [*Sulla-Menashe and Friedl* 2018]) are used to provide
monthly mean land surface emissivities for the mid-wave to thermal IR bands ($3.6 \sim 14.3$ $\mu$m) and
surface white sky albedo for the VNIR bands ($0.4 \sim 2.3$ $\mu$m), respectively, at a $0.05 \times 0.05°$ spatial
resolution. Surface types and snow/sea ice coverage data are from the International Geosphere-
Biosphere Programme (IGBP) and daily Near-real-time Ice and Snow Extent (NISE) data [*Brodzik*
*and Stewart*, 2016], respectively.
**3.3 Clear and cloud phase classifications from existing VIIRS and MODIS products**
Since the present RF models are trained with SNPP VIIRS observations, the first priority of
this study is evaluating and comparing the trained RF models with CALIOP and the existing VIIRS



cloud products. Due to the viewing geometry differences between VIIRS and MODIS, it is difficult
to simply attribute the mask/phase differences from the RF models and MODIS products to
algorithms. However, existing cloud mask and phase products from Aqua MODIS are still
compared as a reference in this work.
The Aqua MODIS and SNPP VIIRS CLDMSK (cloud mask) and CLDPROP (cloud top and
optical properties) [*Ackerman et al.*, 2019] products represent NASA's effort to establish a long-
term consistent cloud climate data record, including cloud detection and thermodynamic phase,
across the MODIS and VIIRS observational records. While the CLDMSK and CLDPROP
algorithms share heritage with the standard Collection 6.1 MODIS cloud mask (MYD35) and
cloud top and optical properties (MYD06) algorithms, both algorithms use only a subset of bands
common to both sensors to minimize differences in instrument spectral information content. The
initial Version 1.0 of the CLDMSK and CLDPROP products were publicly released in April 2019;
CLDPROP has since been reprocessed to Version 1.1, which includes a fix to the optical property
thermodynamic phase algorithm, with public release in October 2019.
The CLDMSK and MYD35 algorithms use a variety of band combinations and thresholds
depending on cloud and surface types [*Frey et al.*, 2008; *Ackerman et al.*, 2008]. Meanwhile, the
algorithms use different approaches for daytime (i.e., solar zenith angle less than 85°) and
nighttime pixels. In the CLDMSK and MYD35 algorithms, pixels are categorized into four
categories, namely confident clear, probably clear, probably cloudy, and cloudy. The CLDPROP
and MYD06 algorithms separate cloudy and probably cloudy pixels into liquid water, ice, and
unknown phase categories. Specifically, the MYD06 product includes two cloud phase algorithms:
an IR-Phase algorithm [*Baum et al.*, 2012] that uses observations in four MODIS IR bands for
daytime and nighttime phase classification (hereafter referred to as the MYD06 IR-Phase), and a
daytime-only algorithm designed for the cloud optical properties retrievals [*Marchant et al.*, 2016;
*Platnick et al.*, 2017] that uses VNIR/SWIR, and IR observations (hereafter referred to as the
MYD06 OP-Phase). A notable change for the VIIRS/MODIS CLDPROP algorithm with respect
to the standard MODIS MYD06 algorithm is the replacement of the MYD06 IR-Phase by a NOAA
operational algorithm originally developed for Clouds from AVHRR-Extended (CLAVR-x)
[*Heidinger et al.*, 2012] and now applied to VIIRS and GOES-16/17. This algorithm is used to
provide cloud top properties, including thermodynamic phase (hereafter CLDPROP CT-Phase), in
the absence of $CO_2$ IR gas absorption bands. IR bands are primarily used in the CLDPROP CT-
Phase algorithm, while complementary SWIR bands are used when available. The MYD06 OP-
Phase algorithm, applied to daytime pixels only, is included with only minor alteration (related to
cloud top properties changes) in the VIIRS/MODIS CLDPROP product (hereafter referred to as
the CLDPROP OP-Phase).

Although the MYD06 and CLDPROP OP-Phase products are developed for "cloudy" and

"probably cloudy" pixels from the MYD35 and CLDMSK products, a Clear Sky Restoral (CSR)
algorithm [*Platnick et al.*, 2017] is implemented to remove "false cloudy" pixels from the clear-
sky conservative MYD35 and CLDMSK products. Specifically, the CSR uses a set of spectral and
spatial reflectance variability tests to remove dust, smoke, and strong sunglint pixels that are
erroneously identified as "cloudy" or "probably cloudy" by the MYD35 and CLDMSK products
[*Platnick et al.*, 2017]. One should keep in mind that the CSR algorithm is only applied for the
optical property retrievals. Thus, the MYD35 and CLDMSK, and consequently the MYD06 IR-
Phase and CLDPROP CT-Phase, may have "false cloudy" pixels in comparison with CALIOP,
while the impact on the MYD06 and CLDPROP OP-Phase is reduced due to the CSR algorithm.
The cloud mask and thermodynamic phase products used in this study are summarized in Table 1.
**4. Model training and validation**



Here we discuss the training of the all-day and daytime RF models for different surface types.
Both shortwave (SW) and IR observations will be used in the daytime models while only IR
observations will be used in the all-day models.
**4.1 Surface Types**
RF models are trained for different surface types, defined here by the Collection 6 (C6) MODIS
annual IGBP surface type product (MCD12C1), to improve model performance over a single
general model for all surface types. Although the MCD12C1 product includes up to 18 surface
types, for this work we attempt to reduce the total number of surface types by combining surface
types with similar spectral white sky albedos and emissivities, as suggested by *Thampi et al.*
[2017]. An annual global IGBP surface type map and surface albedo data from the MODIS
MCD12C1 [*Sulla-Menashe and Friedl* 2018], and a UW-Madison monthly global land surface
emissivity database [*Seemann et al.*, 2008], are used to generate the climatology of land surface
white-sky albedo and IR emissivity spectra. The UW-Madison database is derived using input
from the MODIS operational land surface emissivity product MOD11 [*Wan et al.,* 2004] at six
wavelengths located at 3.8, 3.9, 4.0, 8.6, 11, and 12 $\mu$m. A baseline fit method is applied to fill
the spectral gaps and provides a more comprehensive IR emissivity dataset at 10 wavelengths from
3.6 to 14.3 micron for global land surface with a 0.05° spatial resolution [*Seemann et al.*, 2008].
The MODIS MCD12C1 product also provides a white-sky albedo dataset at 0.47, 0.56, 0.66, 0.86,
1.24, 1.64, and 2.13 $\mu$m with a 0.05° spatial resolution [*Sulla-Menashe and Friedl* 2018]. The
means and standard deviations of surface emissivity and white-sky albedo spectra are shown in
Figures 2 a) and 3 a), respectively, for 16 different land surface types generated from the UW-
Madison and MCD12C1 data in 2015. Land surface types with similar IR emissivity and SW
white-sky albedo spectra are grouped to reduce to the total number of land surface types to 6
(forest, cropland, grassland, snow/ice, barren/desert, and shrubland), as shown in Figures 2 (b-f)



and 3 (b-f). Figure 4 shows an example map of the reduced global surface type data generated
from the MCD12C1 product for 2015.
**4.2 Generating Training/Validation Datasets**

The training and validation data are obtained from a 5-year (2013-2017) SNPP VIIRS and

CALIOP collocated dataset. The collected dataset is generated with a strict collocation algorithm
that fully considers the spatial differences between the two instruments and parallax effects, as
described in *Holz et al.* [2008]. The SNPP VIIRS data include L1B calibrated reflectance and
brightness temperatures, and the CALIOP data include the L2 1 km cloud layer products. A strict
three-step quality control process is applied to all collocated pixels to ensure data quality in the
training process. First, VIIRS 750 m pixels that are potentially contaminated by aerosol are
excluded using a threshold of 0.1 column AOD at 532 nm from the CALIOP L2 5 km aerosol layer
product. Second, each aerosol-free pixel is labelled by one of four categories, namely, clear sky
and liquid, ice, and ambiguous clouds using the CALIOP L2 1 km layer product, where the liquid
and ice categories include both single-layered cloud scenes and multi-layered clouds with the same
thermodynamic phases (e.g., ice over ice, liquid over liquid). Furthermore, the pixels with
ambiguous types, including uncertain/unknown cloud phases from CALIOP and/or overlapping
objects belonging to different types (e.g., cirrus over liquid), are discarded. Finally, horizontally
inhomogeneous pixels, determined when the CALIOP 1 km label changes within 5 consecutive
VIIRS pixels, are discarded. Figure 5 shows the global distributions of the 5-year collocated clear
(first row) and cloudy pixels (second row) before and after applying the three-step quality control.
Globally, 50% of total clear pixels are excluded due to contamination of broken-cloud and/or
aerosol. In particular, a large fraction of clear pixels in central Africa, India, and southern China
(Figure 5c) are excluded due to relatively large aerosol optical thicknesses in those regions. About
40% of global cloudy pixels (Figure 5f) are excluded probably due to cloud heterogeneity and





aerosol contamination. Regions with complicated cloud structures (e.g., the Inter Tropical
Convergence Zone) have the minimum selection rates (20%). The remaining data are separated
into a training/testing population which consists of 33.4, 41.0 and 24.6 million pixels for clear sky,
liquid water cloud, and ice cloud from years 2013-2016, respectively, and a validation dataset that
consists of 6.9, 8.3 and 5.0 million pixels of clear-sky, liquid water cloud, ice cloud, respectively
from year 2017.
**4.3 RF model training and configuration**
RF model performance is determined by both its inputs (spectral or other information) and its
configuration ($N_{Tree}$ and $N_{Depth}$). Therefore, extensive testing must be conducted to find the optimal
inputs and configuration. The 4-year collocated VIIRS-CALIOP dataset from 2013 to 2016 after
quality control (see Section 4.2) is used for both training (75%) and testing (25%) purposes. The
testing set, also known as cross-validation set, is used to tune and optimize the RF model
parameters. Here we define an accuracy score to evaluate the overall model performance. The
accuracy score is the ratio of pixels (samples) that the both the CALIOP and RF model have the
same categories to total pixels.
Table 2 provides accuracy scores of the IR-based all-day model trained and tested with
different inputs. It shows that with a fixed RF model configuration ($N_{Tree} = 150$ and $N_{Depth} = 15$),
the RF all-day model with input #4 and #6 have the best overall accuracy scores for all surface
types. Generally, by including surface skin temperature ($T_s$) and geolocation (i.e., latitude and
longitude), the accuracy scores for all surface types increase by 2-3%. The surface emissivity
vector $\varepsilon_s$ is less important, likely because this information is highly correlated to surface type and
geolocation. In this study, input #4 is selected mainly because with similar performance, it requires
less memory and computing resources, and it is quite possible that more uncertainty is introduced
with the use of a surface emissivity vector $\varepsilon_s$ from another retrieval product.



A set of model configurations ($N_{Tree}$ and $N_{Depth}$) are also tested based on the selected input #4.
While the number of trees and the maximum depth of individual trees are important determinants
for RF model performance, the overall accuracy scores for all surface types are less sensitive to
these two model parameters when more than 100 trees and 10 maximum tree depths are used (not
shown here). Therefore, we trained the RF all-day models with input #4 and the model
configuration used in Table 2, i.e., $N_{Tree}$ = 150 and $N_{Depth}$= 15.
Similar tests for the RF daytime model (IR plus NIR and SWIR observations) showed that the
optimal input includes reflectances in the 0.86, 1.24, 1.38, 1.64 and 2.25$\mu$m bands, BTs in the
same 3 IR bands in the all-day model, $T_s$, geolocation, and solar/satellite viewing zenith angles.
The same model configuration used in the all-day model, e.g., 150 trees with the maximum depth
15, is used in the daytime model. The accuracy scores of the RF daytime model are higher than
the RF all-day model by 2-3% over almost all surface types except high-latitude regions covered
by snow and ice, where the daytime model accuracy score is higher by up to 6% than the all-day
model due to the inclusion of the 1.38, 1.64 and 2.25µm SWIR bands.
**4.4 Evaluating the RF Models**
The trained RF all-day and daytime models are validated using collocated CALIOP data in
2017. Existing VIIRS cloud products CLDMSK and CLDPROP (see Table 1) are included for
direct comparison with the RF models and CALIOP reference. Several other products, such as the
MODIS CLDMSK and CLDPROP and standard MYD35 and MYD06, are also included for
comparison although they could be different from the RF models due to other non-algorithm
reasons, such as the VZA and pixel size differences mentioned before.
*4.5.1 Cloud mask*


Cloud mask from the two RF models and VIIRS/MODIS products are first compared with
CALIOP lidar observations. For the two models, a cloudy pixel indicates a predicted label "liquid"
or "ice". Here we define cloudy and clear pixels as "positive" and "negative" events, respectively.
A true positive rate (TPR) and false positive rate (FPR) can then be used to evaluate model
performance. The TPR and FPR are defined as:
$$\mathrm{TPR} = \frac{TP}{TP+FN},\qquad\qquad(2)$$
$$\mathrm{FPR} = \frac{FP}{FP+TN},\qquad\qquad(3)$$
where TP (True Positive) and TN (True Negative) are the number of lidar-labeled "cloudy" and
"clear" pixels, respectively, that are correctly detected by the models; whereas FN (False Negative)
and FP (False Positive) are the number of lidar-labeled "cloudy" and "clear" pixels incorrectly
identified by the models. Therefore, TPR, also called model sensitivity, indicates the fraction of
all positive events (i.e., lidar cloudy pixels) that are correctly detected by the models. Similarly,
FPR, also called false alarm rate, indicates the fraction of all negative events (i.e., lidar clear pixels)
that are incorrectly detected as positive (cloudy). TPR and FPR are two critical parameters in
model evaluation. A perfect model is associated with a high TPR (close to 1) and a low FPR (close
to 0).
Figure 6 shows daytime cloud mask TPR-FPR plots from the two RF models and the other
products listed in Table 1. Globally, all products agree well with lidar observations (Figure 6a).
The overall TPRs are higher than 0.94 and FPRs are lower than 0.08. The RF daytime model (red
circle), with a TPR of 0.97 and an FPR of 0.05, is slightly better than the RF all-day model (yellow
circle) and other products. Figure 6b-6h show comparisons over different surface types. It is clear
that the RF daytime model has a robust performance for all surface types. The MODIS MYD35
cloud mask algorithm (black circle) performs best over ocean but has a relatively high FPR (0.22)



over forest and low TPR over snow/ice and barren (0.85) regions. As mentioned in Section 3, the
"false cloudy" pixels from MYD35 and CLDMSK may increase the FPRs correspondingly.
The RF all-day model works fairly well and is comparable to other products for all surface
types regardless of the fact that it only uses three IR window channels from VIIRS while all other
products in the daytime models use VNIR observations. Nighttime (SZA > 85°) cloud mask
comparisons are shown in Figure 7. The overall performances of all operational products decrease
in particular for snow/ice regions. For example, the VIIRS/MODIS CLDMSK products over
snow/ice surface have large fractions of missing "cloudy" pixels (e.g., TPRs < 0.7) and false alarm
rates (FPRs > 0.2) over snow/ice surface. The decrease is more likely explained by the lack of
SWIR bands and the small cloud-snow/ice surface temperature contrast during the nighttime of
summer polar regions. However, the RF all-day model has the best performance for nighttime
pixels, indicating the strong capability of ML based algorithm in capturing hidden spectral features
and optimizing dynamic thresholds of clear and cloudy pixels.
*4.5.2 Cloud thermodynamic phase*
The RF model derived cloud thermodynamic phase products are also compared with CALIOP
lidar and existing VIIRS and MODIS products. For consistent nomenclature, we arbitrarily define
ice clouds and liquid water clouds as "positive" and "negative" events, respectively. To focus on
cloud thermodynamic phase classification, pixels detected as "clear" by either the lidar reference
labels or by the RF models and existing products are excluded. The OP-Phase from both MYD06
and CLDPROP, and the IR-Phase from MYD06, have an "unknown phase" category, which is not
included in the TPR-FPR analysis.
Figure 8 shows daytime cloud phase TPR-FPR plots from the two RF models and the
MODIS/VIIRS products. The two RF models and the MODIS MYD06 OP-Phase are the top 3



phase algorithms for all surface types. The MODIS MYD06 IR-Phase, MODIS/VIIRS CLDPROP
OP-Phase, and CT-Phase have either relatively lower TPRs or higher FPRs over particular surface
types, such as shrubland, snow/ice, and barren regions. Comparisons between nighttime phase
algorithms are shown in Figure 9. For nighttime clouds, the RF all-day model works better than
both CT-Phase and IR-Phase algorithms for all surface types.
Figure 10 shows monthly mean daytime cloud and phase fractions from the VIIRS CLDMSK
and CLDPROP OP-Phase products (top row), and those from the RF daytime model (second row),
in January 2017. For the cloud mask comparison, cloud fractions (CF) from the two products have
similar spatial patterns, while it is also clear that the VIIRS CLDMSK CFs are higher over tropical
oceans by approximately 10% and lower over land by 5% (Figure 10 c). This is consistent with
the cloud mask TPR-FPR analysis shown in Figure 6. Over the tropical ocean, the VIIRS
CLDMSK is more "cloudy", probably due to a fraction of sunglint pixels that are detected as liquid
clouds, leading to a large FPR rate. Another reason for the relatively large cloud fraction (or liquid
water cloud fraction) difference is that in regions covered by "broken" cumulus clouds, and or
clouds with more complicated structures, the inherent viewing geometry differences in the training
datasets may adversely affect the performance of the RF models. For example, CALIOP, with a
nadir viewing geometry may observe clear gaps between two small cloud pieces, while VIIRS,
with an oblique viewing angle, detects broken liquid clouds nearby or high clouds along its long
line-of sight. Comparison between the VIIRS product and the RF daytime model shows more ice
clouds from the RF daytime models over land, which is consistent with the cloud phase TPR-FPR
plots as shown in Figure 8. The RF daytime model may have better performance due to the
consideration of surface type. However, it is also important to notice that due to the lack of
"aerosol" types in current training, in central Africa, the RF models may misidentify elevated



smoke as ice cloudy pixels. For most land surface types except snow/ice, the CLDPROP OP-Phase
has lower TPR rates than the RF daytime models by 0.1, in comparison with the CALIOP.
In addition to the higher CFs over low latitude ocean from the VIIRS CLDMSK product, more
pronounced CF (liquid) differences can be found in northeast and northwest China. Cloud
differences in the two regions are spatially correlated with locations that have heavy aerosol
loadings or snow coverage. For example, heavy aerosol loadings due to pollution in Northeast
China, and a wide land snow coverage in Northwest China are frequently observed in the winter.
The VIIRS CLDMSK may identify pixels with white surface and heavy aerosol loadings as
"cloudy". Some of these pixels are expected to be restored to clear-sky category in the CLDPROP
OP-Phase product (Figure 10 f and 10 i). As evidence, Figure 11 shows comparisons between the
VIIRS products and the RF daytime model in July 2017. The large cloud (liquid) fraction
differences over North China vanish in the summer. This indicates that the RF models might be
able to handle complicated (or unexpected) surface type and strong aerosol events better than the
traditional VIIRS algorithm. However, further investigation is required to understand the
performances of both the VIIRS products and the RF models.
**5. Discussion**
In this Section, we will review the strengths and potential limitations and weaknesses of the
RF models.
**5.1 Advantages**
The above results show that the two RF models have better and more consistent performance
over different regions and surface types in comparison with the MODIS and VIIRS products. In
addition to better performance, it is convenient and efficient to apply the present RF models or
other similar ML-based models to other instruments similar to VIIRS, such as the geostationary



imagers Advanced Himawari Imager (AHI) on Himawari-8/9, the ABI on GOES-16/17, and the
Spinning Enhanced Visible and Infrared Imager (SEVIRI) on Meteosat Second Generation, as
long as reliable reference pixel labels are available. The RF models can be trained and tested for
different surface types and using different input variables in a few hours. In contrast, traditional
methods may suffer from the change of instrument, solar/viewing geometries, and surface
conditions. For example, although the MODIS MYD06 OP-Phase and CLDPROP OP-Phase use
similar input and strategies, cloud thermodynamic phases from the two products are different by
up to 5% for all surface pixels, and by up to 20% over surfaces covered by snow/ice (see Figure 8
black and light blue circles). Besides providing a discrete category for each pixel, the RF models
provide an ensemble of predictions and probabilities of individual categories, which are useful
diagnostic variables in evaluating models in complicated scenarios.
**5.2 Limitations and possible caveats**
Although the evaluation demonstrates that the current RF models are highly consistent with
CALIOP, the models may suffer some artifacts due to the quality of the training data and due to
sampling issues.
*5.2.1 Quality of the training data*
The RF models learn spectral structures of cloud/clear pixels according to the reference labels.
As a consequence, the present model performance relies heavily on the quality of CALIOP Level-
2 data. It is already known that the lidar signal has limitations in detecting the bottom of an
optically thick cloud or lower level clouds underneath an opaque cloud [*Sassen and Cho*, 1992].
Cloud phase classification from the RF daytime model, using both SW and IR observations, may
be different from the CALIOP at multi-layer scenes if the top cloud layer is optically thick enough
for lidar (e.g., optical thickness greater than 3). Using combined CALIOP and CloudSat data as



reference and introducing a "multiple layer clouds" category could be a way to mitigate this
impact.
Additional uncertainties may come from the inconsistency in view angles between the
collocated CALIOP labels and VIIRS spectral observations. For instance, CALIOP always has a
quasi-nadir viewing angle (e.g., 3°) whereas the collocated VIIRS observations have a wide VZA
range (e.g., 0° to 50°). A wide VIIRS VZA range in the training dataset improves model
performance, especially for predicting VIIRS pixels with large VZAs. However, the difference
between the CALIOP and VIIRS viewing geometry could create undesirable artifacts in the
training process. As shown in Figure 10, in the descending areas of the Hadley cell over low-
latitude ocean, where marine boundary layer clouds are dominant, there are relatively large CF
differences between the CLDMSK and the RF models. A reason for the large liquid cloud fraction
differences is that the quality of training datasets decreases in regions covered by "broken"
cumulus clouds, and or clouds with more complicated structures. Further investigation is required
to check if the data screening process introduces sampling bias into the training dataset.
*5.2.2 Sampling issue*
Uneven sampling may also influence the training of RF models. Figure 12 shows the cloud
fraction as a function of viewing geometry. Quasi-constant fractions of both liquid and ice clouds
are found for all operational products and the RF models when VZAs are smaller than 45°, except
the MODIS MYD06 IR-Phase, which has a strong VZA dependency. However, liquid (ice) cloud
fractions from the two RF models increase (decrease) rapidly at high VZAs (greater than 50°),
which is likely caused by the sampling issue. A significant fraction of the training data (greater
than 98%) is located in the region with VZA less than 50° (see the gray dashed distributions in
Figure 12). It is difficult to mitigate this issue using collocated VIIRS-CALIOP data or





observations from other similar instruments in the training process. One possible way is using
model- generated synthetic training data and labels with reliable radiative transfer models. Results
from the RF daytime model are not shown in Figure 12 since they are highly consistent with the
RF all-day model.
*5.2.3 Labeling strategy*

For RF or other ML models, each pixel's classification is determined by prediction

probabilities ($P$) of all potential types. Here we selected a regular strategy that labels a pixel using
the class with the highest probability (see Eq. 1). This strategy is logical for problems with two
categories (e.g., cloud mask only). For problems including 3 or more classes, however, the present
strategy is not the only way to label pixels. For example, a pixel is labeled as "clear" if $P_{clear}$ is
larger than both $P_{liquid}$ and $P_{ice}$ according to the current labeling strategy. It is also possible that,
for the same pixel (less than 0.5% for the two RF models), $P_{clear}$ is lower than the sum of $P_{liquid}$
and $P_{ice}$, making a "cloudy" label more appropriate. For the cloud mask and phase problem
discussed in this paper, in addition to pixel labels, users must be aware of probabilities of the three
types. Another possible way to avoid the ambiguous labeling is using two RF models, one for
cloud masking and one for phase, such that a "clear" or "cloudy" label is given first by the cloud
mask model, while a corresponding "liquid" or "ice" label is assigned to "cloudy" pixels in the
cloud phase model. However, two RF models double the training process and require more
computing resources in operational applications. Using multiple RF models becomes more
impractical if both aerosol and cloudy pixels are considered.
**6. Conclusions**

Two Machine-Learning Random Forest (RF) models were trained to provide pixel types (i.e.,

clear, liquid water cloud, and ice cloud) using VIIRS 750-meter spectral observations. A daytime



model that uses NIR, SWIR, and IR bands and an all-day model that only uses IR bands were
trained separately. In the training processes, reference pixel labels are from collocated CALIOP
Level 2 1 km cloud layer and 5 km aerosol layer products from 2013 to 2016. Careful tests were
conducted to optimize model input and configuration. The two RF models were trained for 7
different surface types (i.e., ocean/water, forest, cropland, grassland, snow/ice, barren/desert, and
shrubland) to improve model performance. In addition to geolocation and solar/satellite geometry
information, we found that using 5 NIR and SWIR bands (0.86, 1.24, 1.38, 1.64 and 2.25 $\mu$m) and
three IR bands (8.6, 11, and 12$\mu$m) in the daytime RF model and using the three IR bands and
surface temperatures in the all-day RF model can achieve the best performances for all surface
types.
The cloud mask and thermodynamic phase classifications from the two RF models were
validated using the collocated CALIOP products in 2017. For daytime cloud mask comparisons
over all surface types, the RF daytime model, with a high TPR (0.93 and higher) and low FPR
(0.07 and lower), performs best among all models evaluated, including MODIS MYD35 and
MODIS/VIIRS CLDMSK products. The RF all-day model works fairly well and is comparable to
other products for all surface types, even in daytime when all other products use shortwave
observations and it does not. For the nighttime cloud mask, the RF all-day model has the best
performance over all products, demonstrating the strong capability of ML-based algorithms for
capturing hidden spectral features of clear and cloudy pixels. All nighttime products perform
slightly weaker at snow/ice regions. The decline is likely explained by the lack of SWIR bands
and the small thermal contrast between the clouds and the surface during the summer nighttime in
polar regions. In this case, the ML-based algorithms are not able to compensate for the missing
physical signatures.

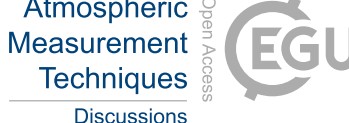

For the daytime cloud thermodynamic phase comparison, we showed that the two RF models
are comparable with the MODIS MYD06 OP-Phase product, and are among the top 3 phase
algorithms for all surface types. The MODIS MYD06 IR-Phase, VIIRS/MODIS CLDPROP OP-
Phase, and CT-Phase have either relatively lower TPRs or higher FPRs over certain surface types,
such as shrubland, snow/ice, and barren regions. For nighttime clouds, the RF all-day model works
better than both CLDPROP CT-Phase and MYD06 IR-Phase for all surface types.
In this study, we have demonstrated the advantages of using ML-based (specifically, RF)
models in cloud masking and thermodynamic phase detections. In contrast to traditional methods,
manually-defined thresholds and matching conditions are no longer needed. The RF models can
be trained and tested for different surface types and using different input variables in a few hours.
Meanwhile, the two RF models show better and more consistent performance over different
regions and surface types in comparison with existing VIIRS and MODIS products. In the future,
more spectral bands and/or spatial patterns can be used to improve pixel classification skills, such
as including more pixel types (e.g., dust and smoke). It is convenient to apply RF models or other
similar ML-based models to other instruments similar to VIIRS with the help of active instruments.
Most importantly, cloud mask and thermodynamic phase products from well-trained RF models
can be used to train other instruments in the absence of active sensors. For example, the current
RF model based VIIRS cloud mask/phase data can be used as reference to train ML-based models
for other instruments, such as MODIS, ABI/AHI, SEVIRI, and airborne instruments.
It is also important to emphasize that the model performance is highly reliant on the quality of
the training samples and reference labels. For example, in this study, more than 98% of the training
data have a VZA less than 50°, leading to an unavoidable bias of cloud phase fractions at large
VZAs. Using synthetic training data generated with reliable radiative transfer models could be a
possible way to mitigate this artifact.



**Acknowledgements**
The authors are grateful for support from the NASA Radiation Sciences Program. C. Wang
acknowledges funding support from NASA through the New (Early Career) Investigator Program
in Earth Science (80NSSC18K0749) managed by Lin Chambers and Allison Leidner. The
computations in this study were performed at the UMBC High Performance Computing Facility
(HPCF). The facility is supported by the U.S. National Science Foundation through the MRI
program (grants CNS-0821258 and CNS-1228778) and the SCREMS program (grant DMS
0821311), with additional substantial support from UMBC. The Collection 6.1 Aqua/MODIS
cloud products (doi: dx.doi.org/10.5067/MODIS/MYD06_L2.061) and MODIS/VIIRS Continuity
cloud products (Version 001) are publicly available from the NASA and Atmosphere Archive and
Distribution System (LAADS) (http://ladsweb.nascom.nasa.gov). The CALIPSO Level 2
Cloud/Aerosol layer products (version 4) products are publicly available from the Atmospheric
Science Data Center (https://eosweb.larc.nasa.gov/).




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





Table 1. Existing VIIRS and MODIS cloud mask and phase products used for comparison. Note
that MYD35 and MYD06 are the standard MODIS Aqua products, and CLDMSK and CLDPROP
are the MODIS Aqua and VIIRS common algorithm continuity products.

| Instrument | Cloud Mask | Cloud Phase |
|---|---|---|
| **MODIS** | MYD35 V6.1 | MYD06 IR-Phase V6.1 |
| | | MYD06 OP-Phase V6.1 |
| | CLDMSK V1.0 | CLDPROP CT-Phase V1.0 |
| | | CLDPROP OP-Phase V1.1 |
| **VIIRS** | CLDMSK V1.0 | CLDPROP CT-Phase V1.0 |
| | | CLDPROP OP-Phase V1.1 |





Table 2: Accuracy scores of RF all-day models based on testing pixels with different inputs and a fixed model
configuration (N_Trees = 150 and Max_TreeDepths = 15).

| # Input | Model Input | Ocean | Forest | Shrubland | Crop | Grassland | Barren | Snow/Ice | All Surface |
|---|---|---|---|---|---|---|---|---|---|
| 1 | $BT_{8.6}$, $BT_{11}$, $BT_{12}$, and VZA | 0.90 | 0.89 | 0.88 | 0.88 | 0.88 | 0.88 | 0.87 | 0.89 |
| 2 | $BT_{8.6}$, $BT_{11}$, $BT_{12}$, VZA, and Lat/Lon | 0.92 | 0.90 | 0.90 | 0.91 | 0.90 | 0.90 | 0.88 | 0.91 |
| 3 | $BT_{8.6}$, $BT_{11}$, $BT_{12}$, VZA, and $T_S$ | 0.93 | 0.91 | 0.90 | 0.91 | 0.90 | 0.90 | 0.89 | 0.92 |
| 4 | $BT_{8.6}$, $BT_{11}$, $BT_{12}$, VZA, Lat/Lon, and $T_S$ | 0.93 | 0.92 | 0.90 | 0.92 | 0.91 | 0.91 | 0.89 | 0.92 |
| 5 | $BT_{8.6}$, $BT_{11}$, $BT_{12}$, VZA, $T_S$, and $\varepsilon_S$ | 0.93 | 0.91 | 0.90 | 0.91 | 0.90 | 0.90 | 0.89 | 0.92 |
| 6 | $BT_{8.6}$, $BT_{11}$, $BT_{12}$, VZA, Lat/Lon, $T_S$, and $\varepsilon_S$ | 0.93 | 0.92 | 0.90 | 0.92 | 0.91 | 0.91 | 0.89 | 0.92 |




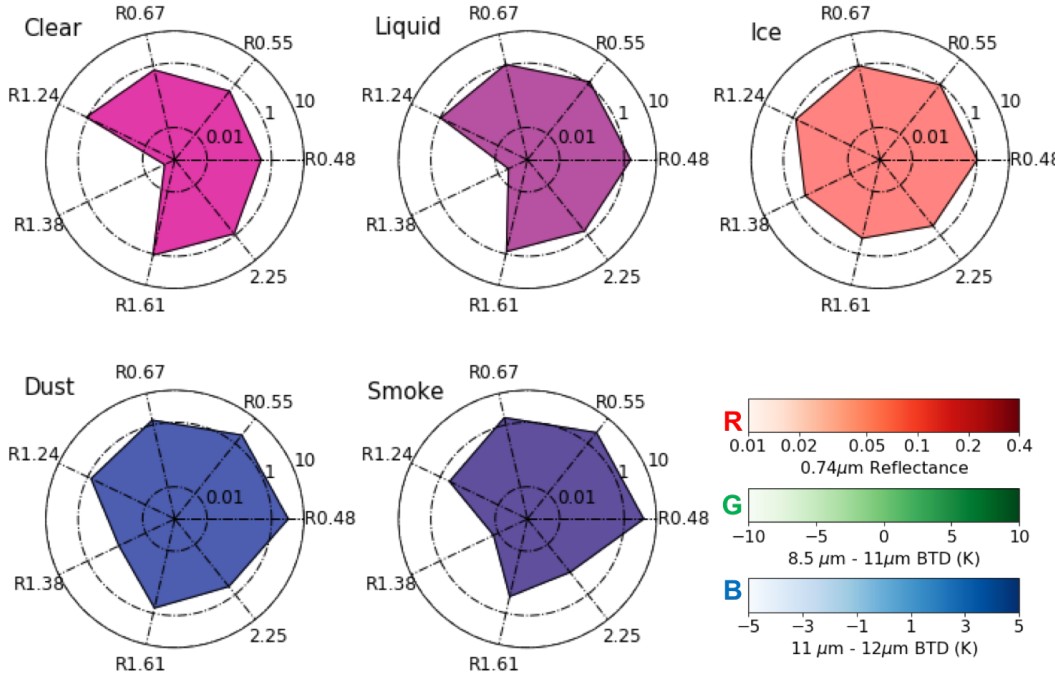

Figure 1. Spectral patterns of the five different pixel types (averaged over 1,000 pixels for each type). For each plot, an apex indicates reflectance ratio between a given VNIR/SWIR band and the 0.86-$\mu$m band, and the spread is filled by false RGB composite (Red: 0.74-$\mu$m reflectance; Green: 8.5-11$\mu$m brightness temperature difference (BTD); Blue: 11-12$\mu$m BTD). The spectral patterns are used in the machine learning algorithms.





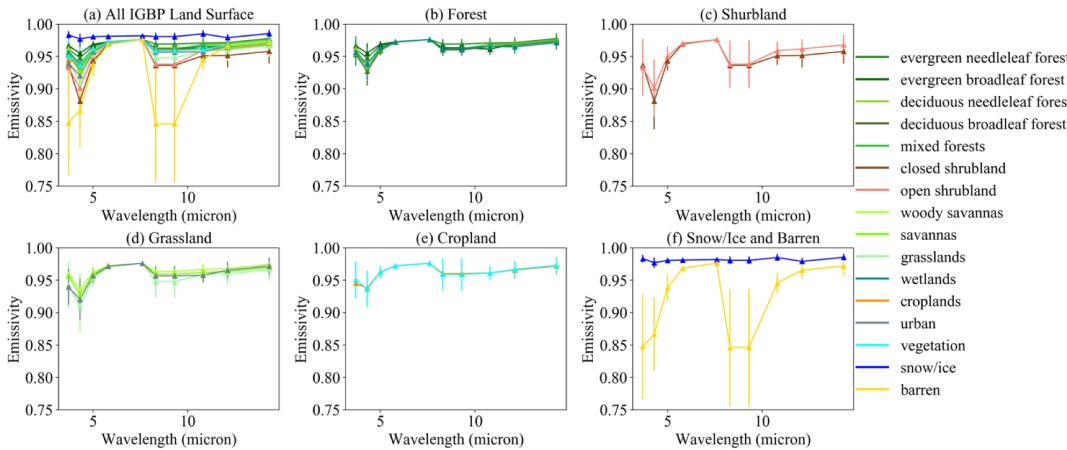

Figure 2. Climatology of the spectral surface emissivity data from the UW-Madison baseline fit
land surface emissivity database [*Seemann et al.*, 2008] for different IGBP surface types. Error
bars indicate the emissivity standard deviations at given wavelengths.





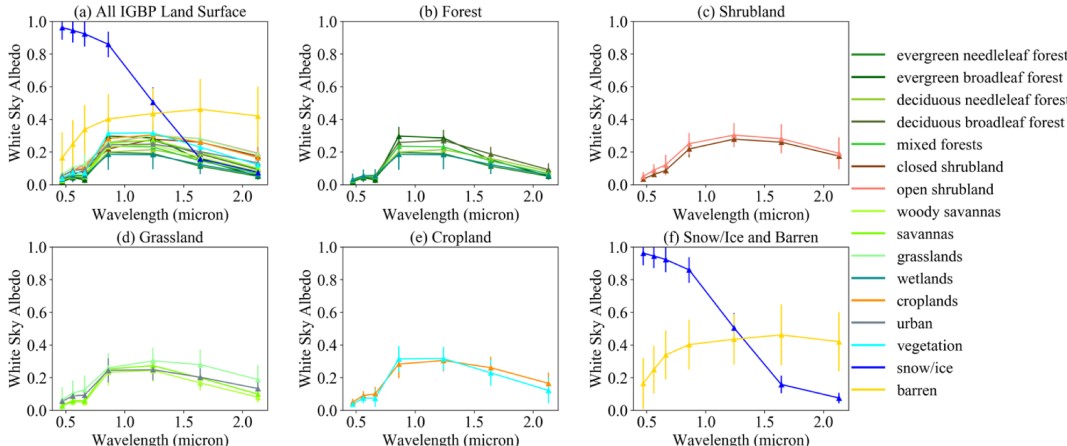

Figure 3. Climatology of the spectral surface white sky surface albedo data from MCD12C1 [*Sulla-*
*Menashe and Friedl* 2018] for different IGBP surface types. Error bars indicate the albedo standard
deviations at given wavelengths.





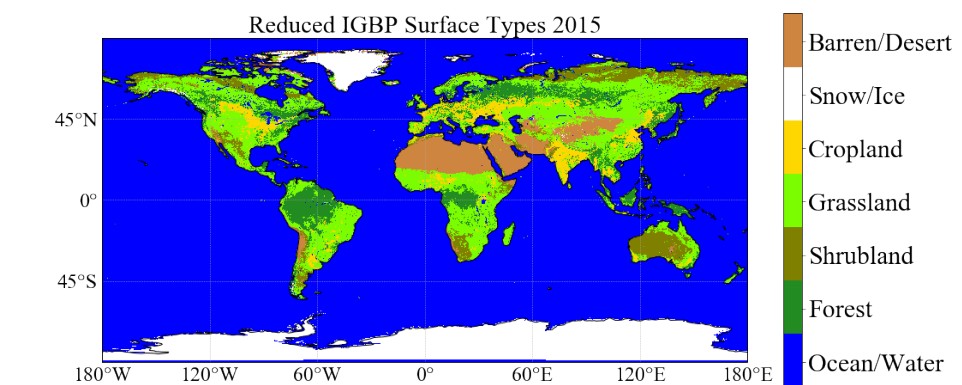


Figure 4. A global map of the seven reduced surface types chosen for the RF model training.





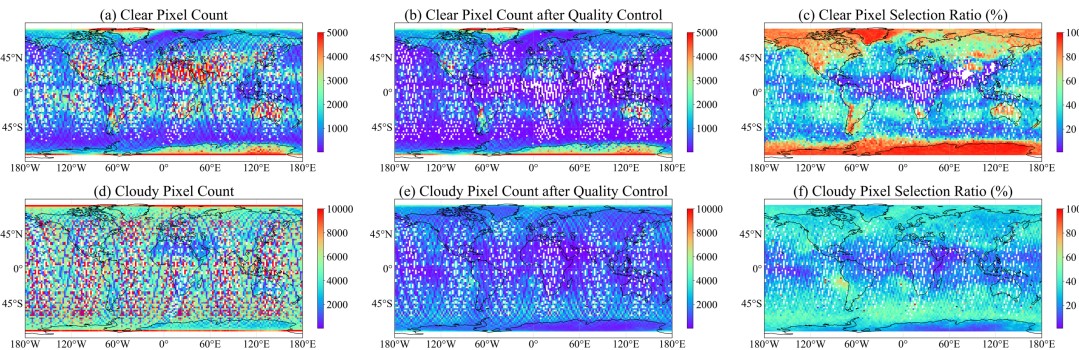

Figure 5. Global distributions of the of clear and cloudy pixels from collocated VIIRS and CALIOP data from 2013 to 2017. Panels a) and d) show the total clear and cloudy pixel counts, respectively. Panels b) and d) show the pixel counts after applying the quality control. The corresponding selection ratios are shown in panels c) and f).

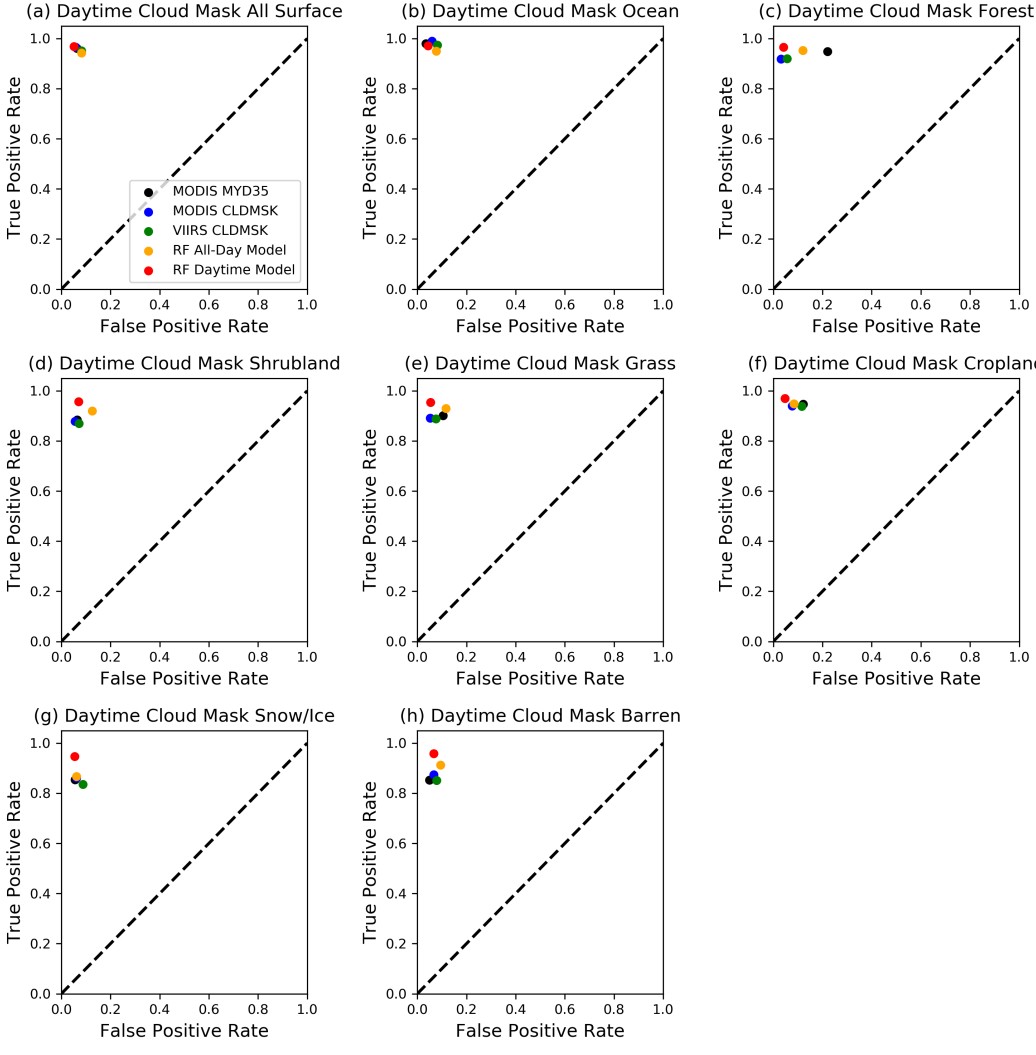

806

Figure 6. False Positive Rate (FPR) versus True Positive Rate (TPR) plots of daytime cloud mask from the two RF models and operational algorithms. Collocated CALIOP Level 2 products in 2017 are used as reference. Global comparisons are shown in panel (a), while panels (b) through (h) show comparisons for difference surface types.


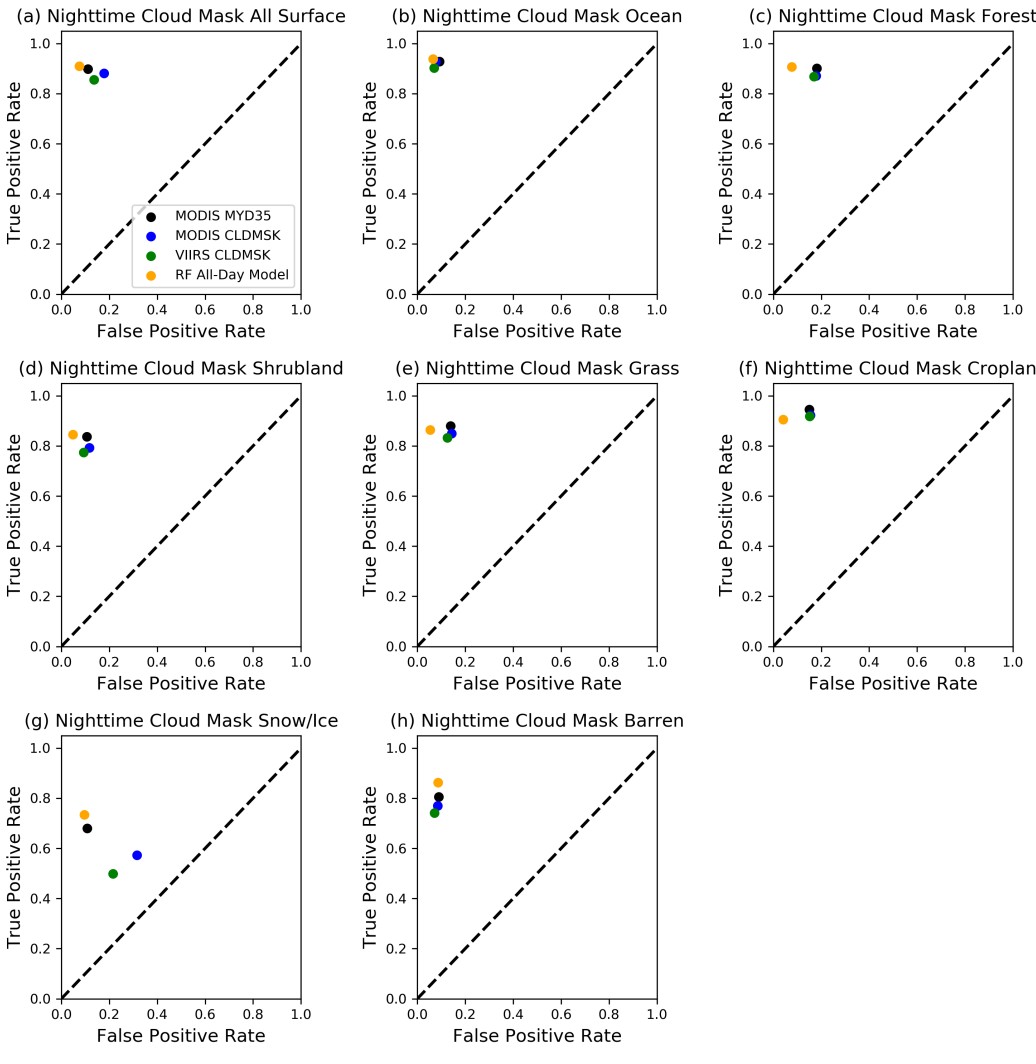


Figure 7. Similar to Figure 6, but for nighttime cloud mask comparisons.






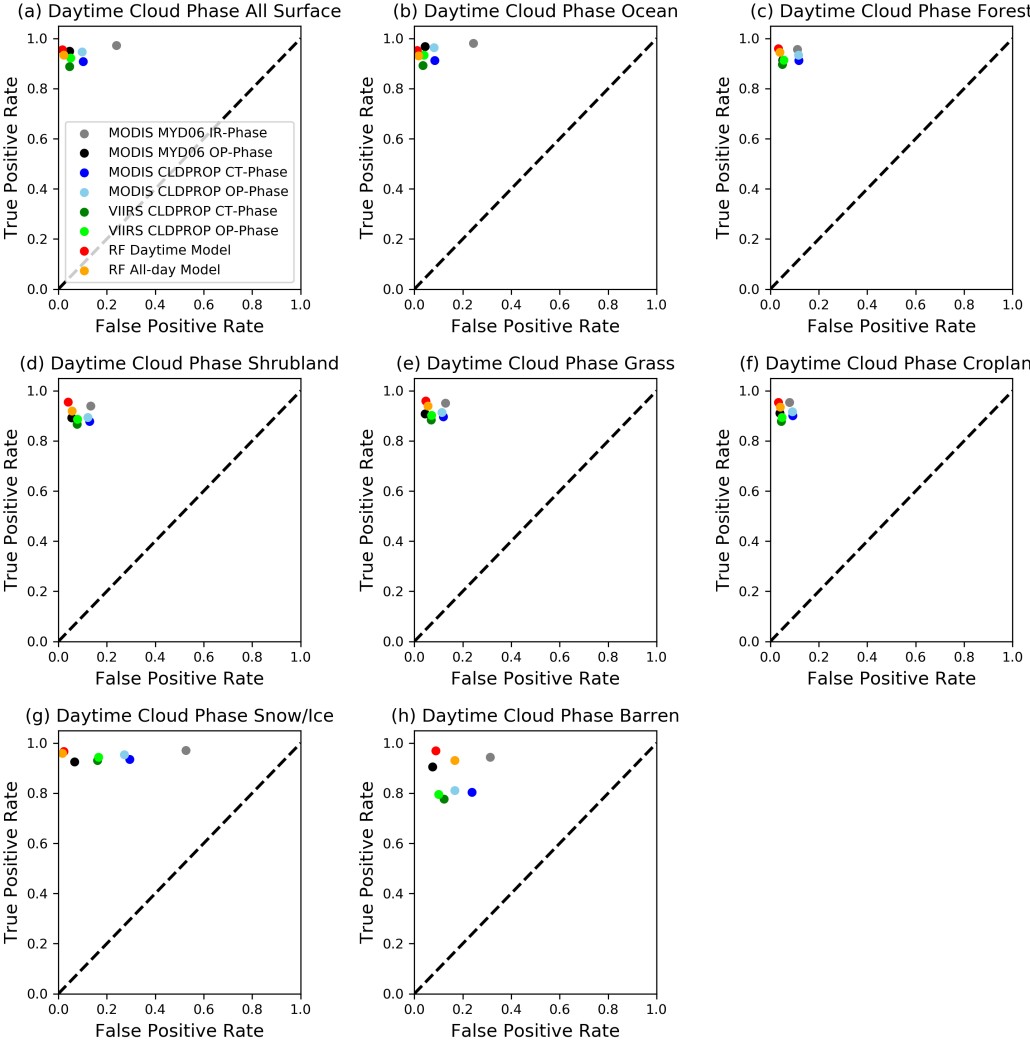


Figure 8. Similar to Figure 6, but for daytime cloud thermodynamic phase comparisons.






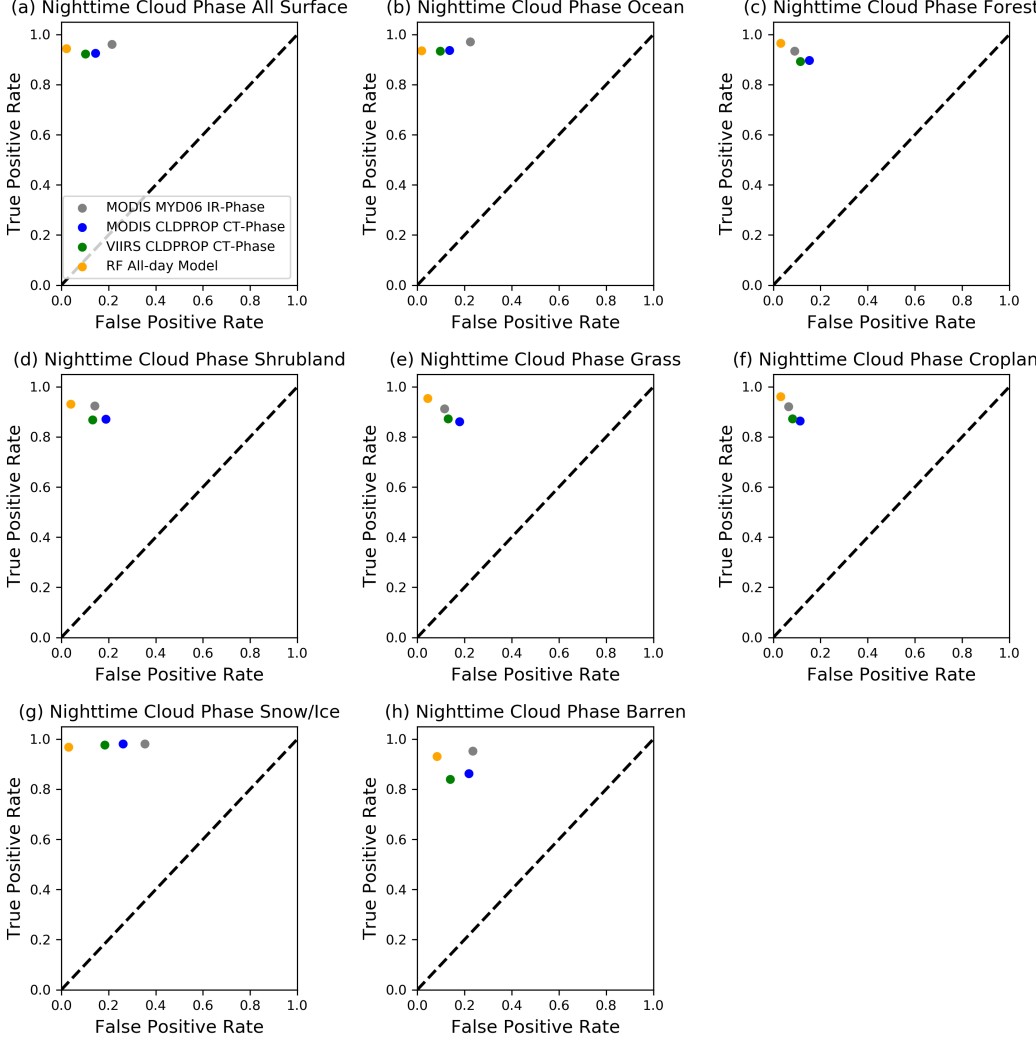


Figure 9. Similar to Figure 6, but for nighttime cloud thermodynamic phase comparisons.

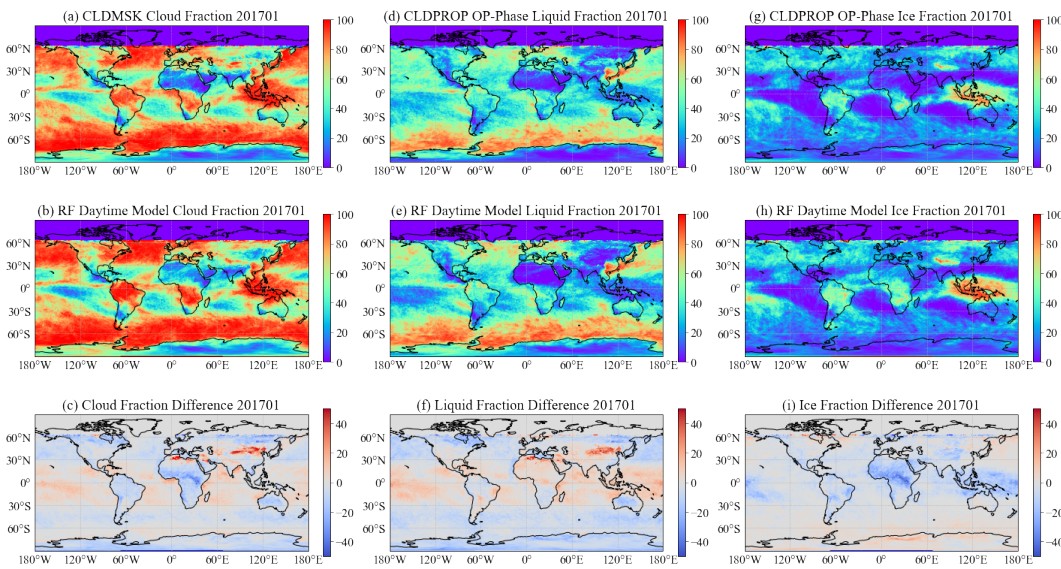


Figure 10. Comparisons between one-month daytime cloud mask and thermodynamic phase
products from the VIIRS CLDMSK and CLDPROP OP-Phase (top row) and the RF daytime
model (second row), and their differences (VIIRS – RF daytime, bottom row) in January, 2017.



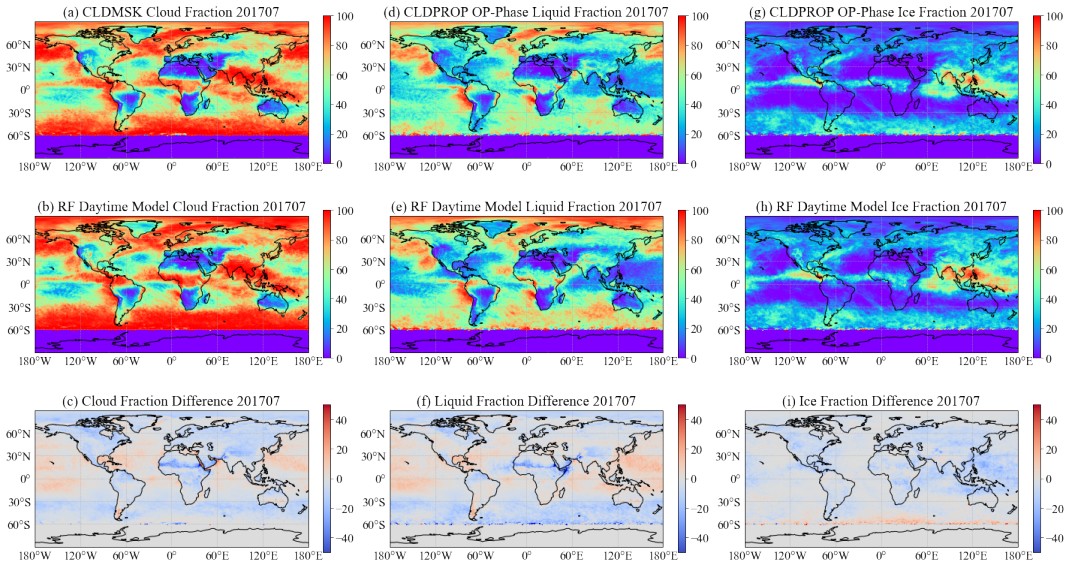

Figure 11. Similar to Figure 10, but for comparisons in July, 2017.






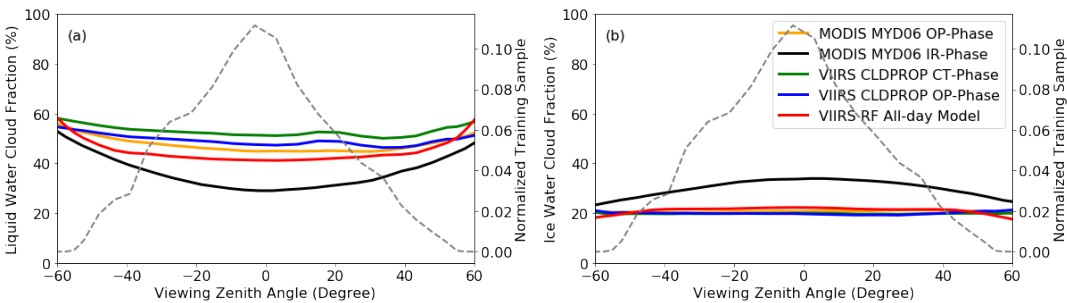

Figure 12. Liquid water (a) and ice (b) cloud fractions as a function of viewing zenith angle from
the one-month daytime cloud mask/phase products in January 2017. The gray dashed curve is the
probability density function of the 4-year VIIRS/CALIOP training samples (2013-2016).