# Peer review of "A Machine Learning-Based Cloud Detection and Thermodynamic Phase Classification Algorithm using Passive Spectral Observations"

_Atmospheric Measurement Techniques, 2019_

## Referee Comment (RC1) · Anonymous Referee #1 · 15 Dec 2019

The authors describe a machine learning (ML) based approach to first detect clouds and second to assign cloud thermodynamic phase (liquid versus ice). The ML algorithm is trained using CALIOP detected liquid and ice clouds but is limited to the most straightforward single phase and single layer cloud configurations (or multilayer with the same phase), thus mixed phase and multi-layered clouds of different phases are not included in this study. The approach is tested against existing MODIS Collection 6 (C6) and MODIS/VIIRS continuity products (both the cloud mask and cloud phase). The ML approach is shown to improve the phase characterization over the existing MODIS

and MODIS/VIIRS continuity algorithms, with greater improvements over certain surface types including snow and ice. Cloud characterization efforts from satellite remote sensing platforms are increasingly utilizing ML algorithms and this paper is timely and a useful exploration of the potential of ML for passive cloud imagery characterization. Parts of the methodology are not as well detailed as need be and the results need to be placed into a broader context. After addressing the comments below and suggestions for straightforward revisions, this paper would be a nice addition to the literature.

Abstract: I found it to be a bit too detailed and meandering. Would suggest tightening it up and focusing on the main points rather than the details.

Line 59: 'having radiometric stability issues' is colloquial and not specific enough to be useful

Lines 79-80: There are two issues here that need to be raised and appear elsewhere. First issue, is this even true? There are many Bayesian methods in the literature that assign uncertainties as a part of the retrieval methodology. Furthermore, using the look up table methodology of MODIS C6, the reported uncertainties for the optical properties appear to be quite useful and rooted in physics. I don't know about uncertainties regarding phase so this could be a different issue. For the cloud mask, the raw Q values are quite useful for an estimate of cloud detection uncertainty. Second issue, calling one set of algorithms 'traditional' is confusing at best. Machine Learning (ML) research dates back to the 1950s and outdates many satellite retrieval algorithm approaches that are currently used. Wording along the lines of "in contrast to most operational and research methods," and similar changes elsewhere, will help make your points clearer. Then you could stick to "ML" as a separate algorithm branch.

Lines 138-139: Now the random forest (RF) model is mentioned and apparently it has a proven record, yet is "not traditional"? The "author classification" of algorithms needs to be reworked throughout the paper.

Lines 233-234: Need to be more explicit as to what "the fix" was to the thermodynamic

phase algorithm.

Section 4.2: This is a long paragraph with a lot of information and should be make clearer than currently written. First recommendation: it would be very helpful to list the pixel count and relative percentages of each of the cloud/no cloud, aerosol/no aerosol, phase and cloud configuration categories that are kept for ML training or are discarded, and should be denoted clearly. It took me a while to figure out that some multilayer clouds are included but only for the same phases. How many multilayer clouds of the same phase occur relative to multiple phases? Second recommendation: say more clearly up front what is in the ML training rather than what is tossed out. Then follow with detail of what is tossed out. It is really hard to keep track of what goes into the sausage. First additional comment: why not attempt to address the ambiguous/mixed phase categories? Some advances in detection and characterization could be made with these types using ML. Do you have plans to do this in follow-on work? Second additional comment: why only use clouds with at least five consecutive labels that are the same? Doesn't this limit the number of cases greatly? Also doesn't this bias the ML training to larger-scale cloud behavior even though the classification is (presumably) done on a pixel-by-pixel basis? Small scale clouds might behave differently (with respect to phase sensitivity) than large scale clouds. Third additional comment: why are cloud structures in the ITCZ any more "complicated" than other geographical regions? What makes a cloud structure "complex"?

Lines 346-353: Is this a description of other experiments tried that are not shown in figures or tables? Or is this paragraph part of the methodology?

Lines 401-405: It would be really helpful to report what total percentage of all pixels considered these represent. The crux of the matter: does ML greatly help for a large percentage of cloudy pixels, or does it help for a small percentage of cloudy pixels? Also, in figures 6-9 showing the true versus false positive rates, it would greatly enhance the presentation of the results by including percentages for each subpanel of the total number of pixels considered.

Lines 418-423: There is a disconnect between this discussion and the earlier discussion on lines 308-310. How are inhomogeneous clouds being considered when earlier the authors state that they are "discarded"? These may be different issues but it is worth making clearer how inhomogeneous clouds are (or are not) considered and dealt with in this study.

Lines 456-457: "A few hours" doesn't really mean anything scientifically. And without describing what is calculated and on what kind of computing platform, this also doesn't convey any information.

Lines 457-459: While not written directly in this way, reading between the lines written by the authors, one could deduce that ML approaches could render instrument calibration efforts and algorithm continuity efforts pointless and irrelevant. Will ML have the potential to address discontinuous satellite observational records by a thorough and accurate labeling of training data for a ML algorithm? I don't think this is what you intended to say, but it does raise the point – can ML methods be used in lieu of a properly calibrated and characterized satellite instrument? Same point applies to lines 467-468.

Lines 470-478: Regarding the use of CALIOP for labeling, one could make the argument that CALIOP is a distinctly different observation and should in fact see something different than a VIS/SWIR observation (e.g., MODIS and VIIRS). Doesn't CALIOP labeling essentially "force" MODIS and VIIRS to observe like a lidar even though they do not contain the same physical sensitivity to clouds as the lidar? Will differences in instrument sensitivity (e.g., CALIOP vs. VIIRS) to a given cloud ultimately lead to poorer performing ML algorithms because one is made to "look like" the other? It is an interesting question to consider. For some clouds, the lidar and passive spectrometer could provide a lot of valuable complementary information, and that is basically "thrown out" in a ML algorithm when one is forced to behave like the other.

Lines 489-490: not sure what is meant by "screening process"

Lines 518-519: why is it more impractical to consider aerosol and cloud together?

---

## Referee Comment (RC2) · Anonymous Referee #2 · 18 Dec 2019

This paper applies a machine learning (ML) approach to the problem of cloud detection and thermodynamic phase assignment from passive satellite measurements. This is potentially significant considering the challenges noted in the manuscript with the traditional methods currently being employed and the rapidly increasing interest in using ML for satellite analyses of clouds. The ML approach evaluates a number of models that are tested and evaluated using various combinations of passive sensor radiances and ancillary data products as inputs while CALIOP data are used to define the reference labels for cloud occurrence and phase. Two models are selected for evaluation,

one that employs solar and infrared radiances (daytime) and one that employs only infrared radiances (all day). The view angle, latitude, longitude and the surface skin temperature were found to be the most important ancillary data needed. In addition, the models are trained for 7 surface types. The two models are found to perform reasonably well and performance metrics generally exceed the current approaches employed on MODIS and VIIRS by the MODIS Science Team. However, the significance of the results are difficult to gauge for a variety of reasons. For example, the ML and current (referred to as traditional in the manuscript) approaches are designed much differently with regards to the targeted clouds, atmospheric correction, scene type dependencies and other factors. With respect to the clouds, the ML model development excludes the most difficult clouds which are pervasive over the Earth. In particular, clouds in polluted environments, broken clouds and single-layer and multi-layer ice overlapping water clouds are screened out of the training and validation dataset. The rationale for taking this approach is not well described. Part of the evaluation of the ML method against current methods with respect to CALIPSO (figs 6-9) could perhaps be considered an apples to apples comparison in that the same pixels are being evaluated. But, considering that the ML approach was developed using a particular subset of (screened) data while the current approaches were designed for application over a much wider range of conditions is possibly unfair, and the comparison are potentially misleading. I wish the authors had taken a more globally applicable ML approach to the problem. It seems to me that at best the results suggest that ML methods can perhaps perform at least as well as the current non-ML methods and that these can be developed for application to other satellites much easier (and cheaper). Despite all of these issues, the study is a reasonable initial step, the results are clearly presented and the manuscript is grammatically clean. Therefore, I find that the manuscript could be published after some revision. In particular, I recommend that the authors clarify the rationale for the approach, clarify the significance of the results, and temper suggestions regarding the potential for ML to improve the accuracies of global cloud analyses since in my view this is not adequately demonstrated here given the heavily restricted

dataset that is used. Additional recommendations to improve the manuscript follow:

Line 23: Strongly suggest something like this: "It is shown using a conservative screening process that excludes the most challenging cloudy pixels for passive remote sensing, . . .

Line 35: 'will' need further attention

Line 62: Zhou reference may need updating

Line 79-80: This statement is too vague and possibly misleading. How is the uncertainty assessment more difficult for a cloud classification derived with the traditional methods vs the ML approach? It is true that in a Bayesian context, uncertainties in satellite retrievals associated with inversion are easy to extract, but these do not include uncertainties w.r.t ground truth data due to simplifying assumptions in the forward models and a host of other factors. Please elaborate to clarify and support your contention.

Line 195: should be Sayer et al 2017?

Line 221-223: not clear what you mean here

Line 231-234. Not sure what the relevance of this update is to the paper unless you used the older version. If this is the case, then you'll need to elaborate on the impact of the deficient version 1.0 algorithm on this study.

Line 249. Not sure what GOES-16/17 have to do with anything. Suggest 'which is now applied to VIIRS.'

Line 301-311: This is an important section with no rationalization for the decisions made to create the training/validation datasets. You should explain why each of these decisions were made and justified

Line 316: define complicated

[Figure]

Line 327: describe how the tuning and optimization were achieved

Line 334: It would be useful to elaborate on possible reasons for the importance of geolocation as an input and the lack of importance for Ts. Why use Ts instead of Tclr computed at TOA? Wouldn't the latter be more consistent with the traditional approaches?

Line 346: Not clear what you mean by similar tests. Consider elaborating further.

Line 348: change to 'IR bands used in the all-day model'

Line 353: Consider tabulatin the daytime results similar to table 2. I think this would be useful.

Line 378 and further: Figs 6-9 are fine but it would help the reader better understand the comparisons if these data could also be tabulated (unless of course you don't think that they are significant enough to further illuminate)

Line 387-389: Is this any surprise considering that you have eliminated the most difficult clouds?

Lines 406-412: the results in figures 8 and 9 are not very clear or well described. In a relative sense, which algorithms are overdetecting or underdetecting ice and water clouds and why?

Line 450: change to something like this "The above results indicate that for the screened data considered here, the two RF models have better and more consistent performance over different regions and surface types in comparison with the MODIS and VIIRS products suggesting the potential to improve the overall performance in more global operational applications

Line 457: It is good to drive home the point regarding the ease and cost savings of applying ML vs the traditional approaches which took years to develop. 'a few hours' seems vague tho. Consider elaborating further.

[Figure]

Line 459-462: Do they really use similar input? The channel complements are different, so if this in any way affects the phase determination, then what you are saying could be unfair and misleading since the two methods were not designed for continuity.

Line 465: In this section it should be emphasized again that a screened dataset is used to train and test the ML methods that excludes the more difficult pixels for passive sensor methods. While the ML methods appear to offer some advantages, the higher accuracies found here compared to the traditional approaches may not be representative of those found when applied to a more inclusive dataset

Lines 474-478: This is also vague and won't make much sense to most readers. What is the objective for your passive determination? Consider elaborating further on the definition and applications for cloud phase (cloud top or radiative), and the relative sensitivities of passive vs active. Maybe then it would be more clear what you mean when you say that a multi-layer clouds category could help

Lines 489-490. The screening process almost certainly impacts the comparisons with the traditional methods which were not developed with a similar screening process. Please make sure that you address this somewhere in the manuscript.

Line 518: why is this more impractical? It actually seems necessary

Line 534: using the collocated CALIOP products in 2017 and excluding the more difficult pixels associated with polluted, broken and mixed-phase cloud conditions

Line 553: should read " . . .phase detections in a limited set of conditions

Line 555: consider changing 'a few hours' to 'considerably more efficiently' ??

Line 562 and 563: change 'can' to 'could'

Line 564: Suggest adding this at the end: It remains as future work to determine how such an approach might lead to improved consistency in cloud properties derived from different satellite remote sensors

Line 607: reformat with last name first or change reference on line 150

Line 651: reformat with last name first or change reference on line 121

Line 829: Why is MODIS CLDPROP not shown in figure 12?

---

## Short Comment (SC1) · 16 Jan 2020

Dear Authors,

I appreciate your work very much and think that this is a valuable contribution to remote sensing. Nevertheless, I think that you also should mention two of our papers in this same journal since they use similar methods of machine learning to perform cloud detection and cloud property derivation. In particular, they also use measurements of the CALIOP lidar as a reference and collocate them with passive observations. You

could also check references therein for papers with similar topics.

Strandgren, J., Bugliaro, L., Sehnke, F., and Schröder, L.: Cirrus cloud retrieval with MSG/SEVIRI using artificial neural networks, Atmos. Meas. Tech., 10, 3547–3573, https://doi.org/10.5194/amt-10-3547-2017, 2017.

Kox, S., Bugliaro, L., and Ostler, A.: Retrieval of cirrus cloud optical thickness and top altitude from geostationary remote sensing, Atmos. Meas. Tech., 7, 3233–3246, https://doi.org/10.5194/amt-7-3233-2014, 2014.

Best regards

–Luca Bugliaro

---

## Author Comment (AC1) · 21 Feb 2020

**Responses to Reviewers**

This document includes our responses to the reviewers' comments and suggestions for the manuscript [doi:10.5194/amt-2019-409]: "A Machine Learning-Based Cloud Detection and Thermodynamic Phase Classification Algorithm using Passive Spectral Observations".

We thank all the reviewers for their helpful suggestions and comments. We hope the revisions are found responsive and appropriate, and that the revised manuscript will be deemed acceptable for publication in the *Atmospheric Measurement Techniques*.

Our responses to the general comments and suggestions from the reviewers (Reviewer #1: Blue; Reviewer #2: Green; Reviewer #3: Orange) are listed below (response in black):

**General Responses:**

R1: The authors describe a machine learning (ML) based approach to first detect clouds and second to assign cloud thermodynamic phase (liquid versus ice). The ML algorithm is trained using CALIOP detected liquid and ice clouds but is limited to the most straightforward single phase and single layer cloud configurations (or multilayer with the same phase), thus mixed phase and multi-layered clouds of different phases are not included in this study. The approach is tested against existing MODIS Collection 6 (C6) and MODIS/VIIRS continuity products (both the cloud mask and cloud phase). The ML approach is shown to improve the phase characterization over the existing MODIS and MODIS/VIIRS continuity algorithms, with greater improvements over certain surface types including snow and ice. Cloud characterization efforts from satellite remote sensing platforms are increasingly utilizing ML algorithms and this paper is timely and a useful exploration of the potential of ML for passive cloud imagery characterization. Parts of the methodology are not as well detailed as need be and the results need to be placed into a broader context. After addressing the comments below and suggestions for straightforward revisions, this paper would be a nice addition to the literature.

Response: We appreciate the insightful comments from the first reviewer (R1). We also noticed that some details, in particular the training/validating dataset selection and model configurations are not well described in the original version. Therefore, in the revised version, we provided more details of the method and results. Please check the new Tables 2-5, and corresponding responses to R1.6, R1.8, and R2.18.

R2: This paper applies a machine learning (ML) approach to the problem of cloud detection and thermodynamic phase assignment from passive satellite measurements. This is potentially significant considering the challenges noted in the manuscript with the traditional methods currently being employed and the rapidly increasing interest in using ML for satellite analyses of clouds. The ML approach evaluates a number of models that are tested and evaluated using various combinations of passive sensor radiances and ancillary data products as inputs while CALIOP data are used to define the reference labels for cloud occurrence and phase. Two models are selected for evaluation, one that employs solar and infrared radiances (daytime) and one that employs only

infrared radiances (all day). The view angle, latitude, longitude and the surface skin temperature were found to be the most important ancillary data needed. In addition, the models are trained for 7 surface types. The two models are found to perform reasonably well and performance metrics generally exceed the current approaches employed on MODIS and VIIRS by the MODIS Science Team. However, the significance of the results are difficult to gauge for a variety of reasons. For example, the ML and current (referred to as traditional in the manuscript) approaches are designed much differently with regards to the targeted clouds, atmospheric correction, scene type dependencies and other factors. With respect to the clouds, the ML model development excludes the most difficult clouds which are pervasive over the Earth. In particular, clouds in polluted environments, broken clouds and single-layer and multi-layer ice overlapping water clouds are screened out of the training and validation dataset. The rationale for taking this approach is not well described. Part of the evaluation of the ML method against current methods with respect to CALIPSO (figs 6-9) could perhaps be considered an apples to apples comparison in that the same pixels are being evaluated. But, considering that the ML approach was developed using a particular subset of (screened) data while the current approaches were designed for application over a much wider range of conditions is possibly unfair, and the comparison are potentially misleading. I wish the authors had taken a more globally applicable ML approach to the problem. It seems to me that at best the results suggest that ML methods can perhaps perform at least as well as the current non-ML methods and that these can be developed for application to other satellites much easier (and cheaper). Despite all of these issues, the study is a reasonable initial step, the results are clearly presented and the manuscript is grammatically clean. Therefore, I find that the manuscript could be published after some revision. In particular, I recommend that the authors clarify the rationale for the approach, clarify the significance of the results, and temper suggestions regarding the potential for ML to improve the accuracies of global cloud analyses since in my view this is not adequately demonstrated here given the heavily restricted dataset that is used.

Response: We appreciate the insightful comments from the second reviewer (R2). We agree with the major concern from R2 that the current training/validation results could be problematic or cannot represent global clouds considering a large fraction of "mixed phase", "inhomogeneous", or "aerosol contaminated" clouds are excluded. To address this concern and other related questions and comments, we made necessary modifications and gave more explanations in the revised manuscript and response. Please find our detailed responses below, in particular responses to R1.6, R1.8, R1.11, R1.12, R2.4, R2.17, R2.18, R2.23, and R2.27.

R3: Comments from Dr. Luca Bugliaro.

Dear Authors,

I appreciate your work very much and think that this is a valuable contribution to remote sensing. Nevertheless, I think that you also should mention two of our papers in this same journal since they use similar methods of machine learning to perform cloud detection and cloud property derivation. In particular, they also use measurements of the CALIOP lidar as a reference and collocate them with passive observations.

Could also check references therein for papers with similar topics.
Strandgren, J., Bugliaro, L., Sehnke, F., and Schröder, L.: Cirrus cloud retrieval with

MSG/SEVIRI using artificial neural networks, Atmos. Meas. Tech., 10, 3547–3573, https://doi.org/10.5194/amt-10-3547-2017, 2017.
Kox, S., Bugliaro, L., and Ostler, A.: Retrieval of cirrus cloud optical thickness and top altitude from geostationary remote sensing, Atmos. Meas. Tech., 7, 3233–3246, https://doi.org/10.5194/amt-7-3233-2014, 2014.

Response: The two papers match the topic perfectly and should be included in the reference list. We appreciate the comments and suggestions from Dr. Luca Bugliaro.

**Detailed Responses**

R1.1: Abstract: I found it to be a bit too detailed and meandering. Would suggest tightening it up and focusing on the main points rather than the details.

Response: Done. We removed some details about the accuracy rates for the two RF models in cloud mask and phase detections.

R1.2: Line 59: 'having radiometric stability issues' is colloquial and not specific enough to be useful.

Response: Done. We replaced "radiometric stability issues" with "calibration drifts".

R1.3: Lines 79-80: There are two issues here that need to be raised and appear elsewhere. First issue, is this even true? There are many Bayesian methods in the literature that assign uncertainties as a part of the retrieval methodology. Furthermore, using the look up table methodology of MODIS C6, the reported uncertainties for the optical properties appear to be quite useful and rooted in physics. I don't know about uncertainties regarding phase so this could be a different issue. For the cloud mask, the raw Q values are quite useful for an estimate of cloud detection uncertainty. Second issue, calling one set of algorithms 'traditional' is confusing at best. Machine Learning (ML) research dates back to the 1950s and outdates many satellite retrieval algorithm approaches that are currently used. Wording along the lines of "in contrast to most operational and research methods," and similar changes elsewhere, will help make your points clearer. Then you could stick to "ML" as a separate algorithm branch.

Response: The reviewer is quite correct that quantitative uncertainty datasets now accompany the retrieval of continuous variables, e.g., MODIS cloud optical properties. And as the reviewer points out, the MODIS CLDMSK cloud detection algorithm reports a continuous "clear sky confidence" or "Q value", ranging from 0 to 1, for each pixel. Therefore, we decided to remove this statement. We have also made additional modifications to the rest of the manuscript. For the second suggestion, we agree with the reviewer. "Traditional" could lead to unnecessary confusion. Therefore, we changed the word "traditional" to "hand-tuned" throughout the manuscript.

R1.4: Lines 138-139: Now the random forest (RF) model is mentioned and apparently it has a proven record, yet is "not traditional"? The "author classification" of algorithms needs to be reworked throughout the paper.

Response: We have changed the word "traditional" to "hand-tuned". See our previous response.

R1.5: Lines 233-234: Need to be more explicit as to what "the fix" was to the thermodynamic phase algorithm.

Response: We found that our initial phase algorithm implemented in CLDPROP Version 1.0, which is based on the MOD06 Collection 6/6.1 optical property phase algorithm with some modification, omitted a key cold cloud sanity check that led to spurious liquid cloud decisions at the edge of ice clouds. This in turn caused spuriously large liquid cloud fractions and a discontinuity in ice cloud effective radius retrieval statistics. We subsequently implemented a new cold cloud sanity check and reprocessed CLDPROP to Version 1.1. More details about this fix and its impacts can be found in the Product Version 1.1 Change Summary section (Section 1.4) of the CLDPROP User's Guide (https://atmosphere-imager.gsfc.nasa.gov/sites/default/files/ModAtmo/EOSSNPPCloudOpticalPropertyContinuityProductUserGuidev11.pdf) available on the Atmosphere Discipline Team website (https://atmosphere-imager.gsfc.nasa.gov/). However, following the second reviewer's comment, we believe this detail is irrelevant to this paper and have decided to remove this statement from Lines 233-234.

R1.6: Section 4.2: This is a long paragraph with a lot of information and should be make clearer than currently written. First recommendation: it would be very helpful to list the pixel count and relative percentages of each of the cloud/no cloud, aerosol/no aerosol, phase and cloud configuration categories that are kept for ML training or are discarded, and should be denoted clearly. It took me a while to figure out that some multilayer clouds are included but only for the same phases. How many multilayer clouds of the same phase occur relative to multiple phases? Second recommendation: say more clearly up front what is in the ML training rather than what is tossed out. Then follow with detail of what is tossed out. It is really hard to keep track of what goes into the sausage. First additional comment: why not attempt to address the ambiguous/mixed phase categories? Some advances in detection and characterization could be made with these types using ML. Do you have plans to do this in follow-on work? Second additional comment: why only use clouds with at least five consecutive labels that are the same? Doesn't this limit the number of cases greatly? Also doesn't this bias the ML training to larger-scale cloud behavior even though the classification is (presumably) done on a pixel-by-pixel basis? Small scale clouds might behave differently (with respect to phase sensitivity) than large scale clouds. Third additional comment: why are cloud structures in the ITCZ any more "complicated" than other geographical regions? What makes a cloud structure "complex"?

Response: We appreciate the very insightful comments and suggestions. Accordingly, we made necessary modifications in Section 4.2 as listed below:

- First, we added a new table (Table 2) that gives more details about the sample. In this table, it is clear how we select highly reliable datasets by using CALIOP L2 products. For all surface types, approximately 39.3% of all collocated VIIRS 750m pixels are selected for training and testing, while 1/3 of all VIIRS pixels are excluded because of aerosol contamination (e.g., column 532nm AOD > 0.05).

- Second, we reorganized the paragraph by mentioning that only aerosol-free, homogenous clear, and homogenous single-phase cloudy pixels are included in the training/validation datasets. Also, we give clear definitions of "*aerosol-free*", "*homogenous*", and "*single-phase cloud*" in the text and in Table 2.

We should note that the performance of ML models is strongly dependent on the quality of the training dataset. In this study, the two RF models are trained and tested with simple yet highly confident samples collected from 2013 to 2016, with the expectation that the RF models will capture the key spectral features from these simple samples more efficiently. Of course, it is then not surprising that the two models perform well when comparing with CALIOP using similar simple samples from 2017. However, we note that many current operational/research-level phase algorithms, including the MYD06 and CLDPROP optical property phase (OP-Phase) algorithms considered in this study, were also tuned (often by hand) with CALIOP using data filtering strategies similar to those employed here (see, e.g., *Baum et al.*, 2012; *Marchant et al.*, 2016). The better performance of the RF models compared with the operational algorithms, even if only for these simple cases, highlights the advanced capabilities of ML approaches over human tuning to more efficiently identify and effectively utilize spectral information content.

That said, the reviewer raises an important point regarding more complicated cloud scenes. For example, we expect that the RF models may recognize signals from both ice and liquid clouds in overlapping cases when the upper layer cloud is not optically thick in the relevant spectral channels. Of course, this is also the case for current operational phase algorithms (e.g., MYD06, CLDPROP) for which tuning/testing also did not include complicated cloud scenes. Nevertheless, we expect that the classification probabilities that are the output of the RF models can provide important information. For instance, we find that, for simple cases (i.e., homogeneous clear or single-phase cloudy), the probability distributions from the RF all-day model have strong peaks (see Figure 10 a, b, and c in the revised manuscript) close to either 0 or 1. However, for more complicated cases, such as ice over liquid cloud (panel d), the liquid and ice probabilities are more broadly distributed, indicating that the RF all-day model may recognize signals from both liquid and ice and therefore provides ambiguous results. Ambiguous liquid/ice probabilities could be used to define a third, "unknown" phase category, following MYD06 and CLDPROP convention, and also provide a useful quality assurance metric for the downstream cloud optical property retrievals. We also would like to point the reviewer to a manuscript that is relevant to the discussion here: *Marchant et al.* (2020), currently in review, gives a more detailed discussion on MYD06 multilayer cloud detection and the impact on phase detection. We have added this discussion in Section 4.4 and Section 5.

[Figure]

Figure: Clear, liquid, and ice probability distribution functions of the RF all-day model for four lidar pixel categories: (a) CALIOP clear, (b) CALIOP liquid water cloud, (c) CALIOP ice cloud, and (d) CALIOP multiple phases. The multiple phase pixels (d) are not used in model training/validation.

New Reference added:

Marchant, B., Platnick, S., Meyer, K., and Wind, G.: Evaluation of the Aqua MODIS Collection 6.1 multilayer cloud detection algorithm through comparisons with CloudSat CPR and CALIPSO CALIOP products, Atmos. Meas. Tech. Discuss., https://doi.org/10.5194/amt-2019-448, in review, 2020.

- Finally, we mentioned that for some regions, such as the ITCZ, the sample selection rates are low because of the complicated cloud structures. For example, clouds always have very complicated vertical structures (such as multiple layers with difference thermodynamic phases) and strong horizontal heterogeneity due to convection. We modified our previous statement for clarity.

R1.7: Lines 346-353: Is this a description of other experiments tried that are not shown in figures or tables? Or is this paragraph part of the methodology?

Response: For the daytime model, we also tried different input combinations. Another table (Table 4) with all of the details are included in the revised version.

R1.8: Lines 401-405: It would be really helpful to report what total percentage of all pixels considered these represent. The crux of the matter: does ML greatly help for a large percentage of cloudy pixels, or does it help for a small percentage of cloudy pixels? Also, in figures 6-9 showing the true versus false positive rates, it would greatly enhance the presentation of the results by including percentages for each subpanel of the total number of pixels considered.

Response: We agree. In the cloud mask and cloud thermodynamic phase TPR-FPR plots (Figs. 6-9), we have added the total number of pixels for the corresponding surface types. Moreover, we have added the following text and a new table (Table 5) to Section 4.5.2 to demonstrate the importance of "unknown phase" category for each cloud phase product:

*"It is also important to note that the number of pixels used for cloud phase TPR-FPR comparisons in Figures 8 and 9 are different for products that have "unknown phase" categories, namely, MYD06 IR-Phase, MYD06 OP-Phase, and CLDPROP OP-Phase. As shown in Table 5, the MYD06 IR-Phase has a relatively large "unknown phase" phase fraction (15% for all surface*

*types and 34% for snow/ice) in comparison to the OP-Phase products from both MYD06 and CLDPROP, which have 2~3% "unknown phase" fraction approximately*".

R1.9: Lines 418-423: There is a disconnect between this discussion and the earlier discussion on lines 308-310. How are inhomogeneous clouds being considered when earlier the authors state that they are "discarded"? These may be different issues but it is worth making clearer how inhomogeneous clouds are (or are not) considered and dealt with in this study.

Response: This comment is related to R1.6. Please see our response above.

R1.10: Lines 456-457: "A few hours" doesn't really mean anything scientifically. And without describing what is calculated and on what kind of computing platform, this also doesn't convey any information.

Response: Please see our response to R1.11.

R1.11: Lines 457-459: While not written directly in this way, reading between the lines written by the authors, one could deduce that ML approaches could render instrument calibration efforts and algorithm continuity efforts pointless and irrelevant. Will ML have the potential to address discontinuous satellite observational records by a thorough and accurate labeling of training data for a ML algorithm? I don't think this is what you intended to say, but it does raise the point – can ML methods be used in lieu of a properly calibrated and characterized satellite instrument? Same point applies to lines 467-468.

Response: For the first question, we believe that instrument calibration efforts and algorithm continuity efforts are very important. Instead, our main point is that ML approaches have the potential to streamline algorithm tuning and/or threshold selection processes that often occur in response to instrument calibration changes or when porting to other instruments. With non-ML methods, such tuning and/or threshold selection processes need to be done manually, which is a time-consuming effort. We have modified the text in response to the reviewer's comments.

*"With hand-tuned methods, adjustment is always required in the case of calibration changes, algorithm porting to another similar instrument, or changes in solar/viewing geometries and surface conditions. Manual adjustments can be time-consuming (e.g., months or years), whereas the two RF models used in this study were trained and tested for 7 surface types and using different input variables in 3 hours (on an HPC Platform using 32 Intel Xeon Gold 6126 Processors @ 2.60 GHz). More important, manual algorithm adjustment may not provide the best continuity between two instruments. For example, although the MODIS CLDPROP OP-Phase and VIIRS CLDPROP OP-Phase are designed for climate record continuity purpose, cloud thermodynamic phases from the two products are different by up to 4% for all surface pixels, and by up to 10% over surfaces covered by snow/ice (see Figure 8 light blue and light green dots). Further investigation is necessary to understand if, using ML approaches, a better climate record continuity will be achieved with a uniform training dataset.*"

For the reviewer's second question, it is likely true that a properly trained ML algorithm can still achieve a high level of skill in the presence of calibration errors if (a) calibration errors are

relatively small and spectrally/spatially uncorrelated in such a way that physically-relevant signals are not masked by the errors/correlations, and (b) the instrument is radiometrically stable or radiometric changes are monitored/corrected on orbit (which gets back to our main point above). Confirmation of both assumptions requires a dedicated and robust on orbit instrument characterization effort.

R1.12: Lines 470-478: Regarding the use of CALIOP for labeling, one could make the argument that CALIOP is a distinctly different observation and should in fact see something different than a VIS/SWIR observation (e.g., MODIS and VIIRS). Doesn't CALIOP labeling essentially "force" MODIS and VIIRS to observe like a lidar even though they do not contain the same physical sensitivity to clouds as the lidar? Will differences in instrument sensitivity (e.g., CALIOP vs. VIIRS) to a given cloud ultimately lead to poorer performing ML algorithms because one is made to "look like" the other? It is an interesting question to consider. For some clouds, the lidar and passive spectrometer could provide a lot of valuable complementary information, and that is basically "thrown out" in a ML algorithm when one is forced to behave like the other.

Response: We agree with the reviewer's comment regarding different sensitivities between MODIS/VIIRS and CALIOP. This in fact is the reason why we only train the models with simple, single-phase samples for which we expect agreement between the passive and active sensors. This allows the models to learn the spectral signatures of liquid and ice clouds separately. For more complicated cases, i.e., horizontally/vertically heterogeneous and/or multilayer pixels, we then let the models make their own decisions regarding what phase makes the most radiative sense given the observations. Further discussion can be found in our response to R1.6.

R1.13: Lines 489-490: not sure what is meant by "screening process"

Response: We modified our statement to "*to check if the training dataset collection process introduces*".

R1.14: Lines 518-519: why is it more impractical to consider aerosol and cloud together?

Response: Adding complexity to the RF (or other ML) model requires more overhead, such as memory at run-time, computational resources, etc. It could be a potential (but not critical) problem when implementing in an operational algorithm production environment, where there often are limitations on such resources (e.g., caps on memory usage). That said, we decided to remove this statement because there are ways to mitigate these technical issues given sufficient resources.

R2.1: Line 23: Strongly suggest something like this: "It is shown using a conservative screening process that excludes the most challenging cloudy pixels for passive remote sensing…

Response: Done.

R2.2: Line 35: 'will' need further attention

Response: Corrected.

R2.3: Line 62: Zhou reference may need updating

Response: We removed this reference since this paper is not submitted.

R2.4: Line 79-80: This statement is too vague and possibly misleading. How is the uncertainty assessment more difficult for a cloud classification derived with the traditional methods vs the ML approach? It is true that in a Bayesian context, uncertainties in satellite retrievals associated with inversion are easy to extract, but these do not include uncertainties w.r.t ground truth data due to simplifying assumptions in the forward models and a host of other factors. Please elaborate to clarify and support your contention.

Response: We agree with the reviewer's point. Quantitative uncertainties are available for Bayesian methods, and are frequently used in retrievals of continuous variables, e.g., cloud-top height, cloud optical thickness, etc. Furthermore, in the MODIS CLDMSK cloud detection algorithm, a continuous "clear sky confidence" or "Q value", ranging from 0 to 1, is provided for each pixel. Therefore, we decided to remove this statement. Please also see our response to comment: R1.3.

R2.5: Line 195: should be Sayer et al 2017?

Response: Corrected.

R2.6: Line 221-223: not clear what you mean here.

Response: Thanks for pointing it out. We removed this statement from this paragraph.

R2.7: Line 231-234. Not sure what the relevance of this update is to the paper unless you used the older version. If this is the case, then you'll need to elaborate on the impact of the deficient version 1.0 algorithm on this study.

Response: We agree with the reviewer. We removed this statement because it is irrelevant to this paper. Please also see our response to R1.5.

R2.8: Line 249. Not sure what GOES-16/17 have to do with anything. Suggest 'which is now applied to VIIRS.'

Response: Done.

R2.9: Line 301-311: This is an important section with no rationalization for the decisions made to create the training/validation datasets. You should explain why each of these decisions were made and justified.

R2.10: Line 316: define complicated.

Response (2.9 and 2.10): Thanks for the suggestions. Both are highly relevant to comments from the first reviewer R1.6 and R1.12. We gave a very comprehensive response and made necessary modifications.

R2.11: Line 327: describe how the tuning and optimization were achieved.

Response: The remainder of Section 4.3 gives a brief introduction of the tuning and optimization. However, to make our point more clearly, we have added the following statement to the revised text: "*In this study, we tested six groups of input variables for each RF model. The set of model input variables with a relatively high accuracy score and low memory/computing requirement will be selected.*"

R2.12: Line 334: It would be useful to elaborate on possible reasons for the importance of geolocation as an input and the lack of importance for Ts. Why use Ts instead of Tclr computed at TOA? Wouldn't the latter be more consistent with the traditional approaches?

Response: As shown in Table 3, we found that both geolocation and Ts are important in the RF all-day model. $\varepsilon_s$ is less important likely because it is correlated to surface type and geolocation. Here we use Ts instead of $T_{clr}$ because the calculation of $T_{clr}$ requires more input (e.g., temperature/humidity profiles), and a RT model, which introduces more uncertainty and requires more computational resources.

R2.13: Line 346: Not clear what you mean by similar tests. Consider elaborating further.

Response: We modified the "similar tests" to "similar input variable tests". For the daytime model, we also tried 6 different input combinations. We added another table (Table 4) in the revised version.

R2.14: Line 348: change to 'IR bands used in the all-day model'

Response: Corrected.

R2.15: Line 353: Consider tabulating the daytime results similar to table 2. I think this would be useful.

Response: Done.

R2.16: Line 378 and further: Figs 6-9 are fine but it would help the reader better understand the comparisons if these data could also be tabulated (unless of course you don't think that they are significant enough to further illuminate)

Response: We agree with the reviewer. To make the figures easier to understand, we have added the total number of pixels for each surface type to the corresponding plot. Moreover, we have inserted a detailed description of "unknown phase" category and a new table (Table 5) in Section 4.5.2 to demonstrate the importance of "unknown phase" category for each cloud phase product.

 Line 387-389: Is this any surprise considering that you have eliminated the most difficult clouds?

Response: As mentioned at the beginning of this section (Section 4.4), we emphasized that the comparisons (shown in Figures 6-9) are also based on "aerosol-free", "homogeneous", "single-phase" pixels. It is not a big surprise considering that these simple cases are used in model training and testing (see Tables 3 and 4). However, we were surprised by the performance of the RF all-day model. Although only 3 IR window bands are used, the TPR-FPR points from the RF all-day model looks much better than the current MODIS MYD06 IR-Phase, and are comparable to the OP-Phase that uses more spectral information from shortwave bands.

R2.18: Lines 406-412: the results in figures 8 and 9 are not very clear or well described. In a relative sense, which algorithms are overdetecting or underdetecting ice and water clouds and why?

Response: For cloud phase classification, we arbitrarily define ice clouds and liquid water clouds as "positive" and "negative" events, respectively. Therefore, a low TPR indicates underestimation of ice cloud fraction, while a high FPR indicates a large fraction of liquid water cloud samples are identified as ice cloud. It is found that for snow/ice and barren regions, many non-ML models have much lower accuracy rates than for ocean and grassland surfaces. Possible reasons include strong surface reflection, low surface cloud contrast, relatively less training samples and high solar zenith angles (for snow/ice surface).

To address the reviewer's questions, we have added the following statement to Section 4.5.2:
"*A low TPR indicates underestimation of ice cloud fraction, while a high FPR indicates a large fraction of liquid water cloud samples are identified as ice cloud.*"
"*Overall, the performance of the hand-tuned algorithms decreases significantly over snow/ice or barren surfaces. For example, the TPR-FPR plot shows that over daytime snow/ice surface (Figure 8 g), the MODIS CLDPROP OP-Phase and MODIS MYD06 IR-Phase frequently predict liquid water cloud as ice cloud. Similar to the daytime plot, the MYD06 IR-Phase also shows a high FPR rate over snow/ice surface, indicating an overestimated (underestimated) ice (liquid water) cloud fraction. Possible reasons include strong surface reflection, low surface cloud contrast, relatively less training samples and high solar zenith angles. However, the two RF models work fairly well and show consistent accuracy rates across all surface types.*"

R2.19: Line 450: change to something like this "The above results indicate that for the screened data considered here, the two RF models have better and more consistent performance over different regions and surface types in comparison with the MODIS and VIIRS products suggesting the potential to improve the overall performance in more global operational applications.

Response: Done. We appreciate the reviewer's suggestion.

R2.20: Line 457: It is good to drive home the point regarding the ease and cost savings of applying ML vs the traditional approaches which took years to develop. 'a few hours' seems vague tho. Consider elaborating further.

Response: Good point! We reorganized the structure of this paragraph by including necessary information on the "labor comparison" between ML and non-ML methods. Please also see our response to R1.11 for more details.

R2.21: Line 459-462: Do they really use similar input? The channel complements are different, so if this in any way affects the phase determination, then what you are saying could be unfair and misleading since the two methods were not designed for continuity.

Response: We modified the statement to "For example, although the MODIS CLDPROP OP-Phase and VIIRS CLDPROP OP-Phase are designed for climate record continuity purpose, cloud thermodynamic phases from the two products are different by up to 4% for all surface pixels, and by up to 10% over surfaces covered by snow/ice (see Figure 8 light blue and light green dots)."

R2.22: Line 465: In this section it should be emphasized again that a screened dataset is used to train and test the ML methods that excludes the more difficult pixels for passive sensor methods. While the ML methods appear to offer some advantages, the higher accuracies found here compared to the traditional approaches may not be representative of those found when applied to a more inclusive dataset.

Response: We agree with the reviewer, though we note that the traditional approaches considered in this study, particularly the MYD06 and CLDPROP OP-Phase algorithms, were themselves tuned off of CALIOP data using similar single-phase data screening (see *Marchant et al.*, 2016), and thus may also suffer degraded performance in complex scenes. In the revised version, we have added a new paragraph and a new figure to demonstrate the performance of the RF all-day model with CALIOP detected multi-phase scenes. We find that probabilities could be more informative than using a single "label". It is obvious that for complicated samples, ice/liquid cloud probabilities from the RF model are more broadly distributed, resulting in a reduced peak at either 0 or 1. However, further investigation is required to understand how to quantitatively use these probabilities in complex cases. Please also see our response to R1.6.

R2.23: Lines 474-478: This is also vague and won't make much sense to most readers. What is the objective for your passive determination? Consider elaborating further on the definition and applications for cloud phase (cloud top or radiative), and the relative sensitivities of passive vs active. Maybe then it would be more clear what you mean when you say that a multi-layer clouds category could help.

Response: We agree with the reviewer. In this section, our intent is to mention the limitations of using CALIOP data only for the collection of "simple" cases. Therefore, we modified this paragraph as:

"*The RF models learn spectral structures of cloud/clear pixels according to the reference labels. As a consequence, the present model performance relies heavily on the quality of CALIOP Level-2 data. It is already known that the lidar signal has limitations in detecting the bottom of an optically thick cloud or lower level clouds underneath an opaque cloud [Sassen and Cho, 1992]. Some complicated multiple-phase scenes may be misidentified as simple single-phase scenes due*

*to the penetration limit of CALIOP (e.g., the uppermost ice cloud optical thickness greater than 3). Using combined CALIOP and CloudSat data as reference in the future could be a better way to improve the training/validation datasets [Marchant et al., 2020]. However, as noted in that study, CloudSat observations cannot be used without careful filtering since a multilayer scene that is radiatively indistinct from the upper level cloud layer is not necessarily consistent with multilayer detection detected from a cloud radar.*"

R2.24: Lines 489-490. The screening process almost certainly impacts the comparisons with the traditional methods which were not developed with a similar screening process. Please make sure that you address this somewhere in the manuscript.

Response: The non-ML approaches considered in this study, particularly the MYD06 and CLDPROP OP-Phase algorithms, use a similar data screening (see *Marchant et al.*, 2016), and thus may also suffer degraded performance in complex scenes. It is very hard to quantitatively estimate to what extent the screening process could impact those non-ML methods. However, in the revised version, we provided more details about the data selection strategy in Section 4.2 plus two new Tables (2 and 5).

R2.25: Line 518: why is this more impractical? It actually seems necessary.

Response: Adding complexity to the RF (or other ML) model requires more overhead, such as memory at run-time, computational resources, etc. It could be a potential (but not critical) problem when implementing in an operational algorithm production environment, where there often are limitations on such resources (e.g., caps on memory usage). That said, we decided to remove this statement because there are ways to mitigate these technical issues given sufficient resources.

R2.26: Line 534: using the collocated CALIOP products in 2017 and excluding the more difficult pixels associated with polluted, broken and mixed-phase cloud conditions.

Response: Corrected.

R2.27: Line 553: should read " : : :phase detections in a limited set of conditions.

Response: We understand the reviewer's concern. Instead of simply adding "in a limited set of conditions" here, we updated this paragraph to:

"*In this study, we have demonstrated the advantages of using ML-based (specifically, RF) models in cloud masking and thermodynamic phase detection. In contrast with hand-tuned methods, the RF models can be efficiently trained and tested for different surface types and using different input variables. Meanwhile, for aerosol-free, homogeneous samples, the two RF models show better and more consistent performance over different regions and surface types in comparison with existing VIIRS and MODIS datasets. For more complicated scenes, RF probabilities are more informative than binary mask/phase designations. However, further investigation is required to understand how to use probabilities more quantitatively.*"

R2.28: Line 555: consider changing 'a few hours' to 'considerably more efficiently' ??

Response: Done.

R2.29: Line 562 and 563: change 'can' to 'could'

Response: Done.

R2.30: Line 564: Suggest adding this at the end: It remains as future work to determine how such an approach might lead to improved consistency in cloud properties derived from different satellite remote sensors.

Response: Done.

R2.31: Line 607: reformat with last name first or change reference on line 150.

Response: Done.

R2.32: Line 651: reformat with last name first or change reference on line 121

Response: Done.

R2.33: Line 829: Why is MODIS CLDPROP not shown in figure 12?

Response: For legibility reasons, we decided to limit the number of line plots in the figure. The MODIS CLDPROP curves are not included because their locations and structures are quite similar to the VIIRS products.